# SEMDICE: Off-policy State Entropy Maximization via Stationary Distribution Correction Estimation

**Jongmin Lee**[*,1,2]   **Meiqi Sun**[*,1]   **Pieter Abbeel**[1]
[1]UC Berkeley   [2]Yonsei University

## Abstract

In the unsupervised pre-training for reinforcement learning, the agent aims to learn a prior policy for downstream tasks without relying on task-specific reward functions. We focus on state entropy maximization (SEM), where the goal is to learn a policy that maximizes the entropy of the state's stationary distribution. In this paper, we introduce SEMDICE, a principled off-policy algorithm that computes an SEM policy from an arbitrary off-policy dataset, which optimizes the policy directly within the space of stationary distributions. SEMDICE computes a single, stationary Markov state-entropy-maximizing policy from an arbitrary off-policy dataset. Experimental results demonstrate that SEMDICE outperforms baseline algorithms in maximizing state entropy while achieving the best adaptation efficiency for downstream tasks among SEM-based unsupervised RL pre-training methods.

## 1 Introduction

The essence of intelligent agents lies in their ability to learn from experiences. Reinforcement learning (RL) (Sutton & Barto, 1998) provides a framework for autonomously acquiring an intelligent behavior by interacting with the environment while receiving reward signals, and it has shown great promise in various domains such as complex games (Mnih et al., 2015; Silver et al., 2017) and robotic control (Lillicrap et al., 2016; Haarnoja et al., 2018). Still, standard RL algorithms learn *tabula rasa*, i.e. learning from scratch for every task to maximize extrinsic rewards without using previously learned knowledge. Consequently, the learned RL policy tends to be brittle and lacks generalization capabilities, limiting its widespread adoption to many real-world sequential decision-making problems (Cobbe et al., 2020; Gleave et al., 2020).

In contrast, in the fields of computer vision (He et al., 2020; Chen et al., 2020) and natural language processing (Devlin et al., 2019; Brown et al., 2020), large-scale unsupervised pre-training has paved the way for sample-efficient few-shot adaptation. During unsupervised pre-training, the models are trained using a large unlabelled dataset, and then the pre-trained models are fine-tuned later for each specific downstream task with a handful of task-specific labeled datasets. We consider the analogy setting of the unsupervised pre-training for RL (Laskin et al., 2021). Specifically, during unsupervised RL pre-training, the agent is allowed to train over long periods without access to the extrinsic rewards of the environment. This procedure yields a pre-trained policy snapshot, aiming for data-efficient adaptation in subsequent downstream tasks defined by reward functions that were inaccessible a priori.

We consider the State-Entropy Maximization (SEM) approach for RL policy pre-training (Hazan et al., 2019; Liu & Abbeel, 2021b; Yarats et al., 2021), as it is simple yet provides robust policy initialization for efficient adaptation against the worst-case reward function (Eysenbach et al., 2022). However, despite its conceptual simplicity and widespread popularity, a principled approach to sample-efficient *off-policy* SEM methods remains unexplored. Existing methods are either on-policy (thus sample-inefficient) (Hazan et al., 2019), or off-policy but biased (Liu & Abbeel, 2021b; Yarats et al., 2021). They construct the intrinsic reward function based on the particle-based entropy

---

*Equal contribution

estimator (Singh et al., 2003), and then optimize the policy in the direction of maximizing the constructed intrinsic rewards. For off-policy learning algorithms, state particles from the replay buffer are naively used to estimate state entropy (Liu & Abbeel, 2021b; Yarats et al., 2021), resulting in estimates of the entropy of the state data stored in the replay buffer, rather than the state entropy of the target policy's state stationary distribution. Consequently, it remains unclear whether these off-policy methods can converge to an optimal SEM policy. One can consider importance sampling to correct the off-policyness (Mutti et al., 2021), but it would suffer from high variance issue due to the curse of horizon (Liu et al., 2018), significantly limiting the degree of sample reuse. Consequently, none of the existing methods serves as an unbiased, sample-efficient SEM method.

In this paper, we present a state-entropy maximization method that precisely addresses the aforementioned issues of the existing methods. Our method, **S**tate-**E**ntropy-**M**aximization via stationary **DI**stribution **C**orrection **E**stimation (SEMDICE), essentially optimizes in the space of stationary distributions, rather than in the space of policies or Q-functions, and it leverages arbitrary off-policy samples. We show that SEMDICE only requires solving a single convex minimization problem, and thus it can be optimized stably. To the best of our knowledge, SEMDICE is the first principled and practical algorithm that computes a SEM policy from **arbitrary off-policy dataset**. In the experiments, we demonstrate that SEMDICE converges to an optimal SEM policy in the tabular MDP experiments. Also, regarding RL policy pre-training, SEMDICE adapts to downstream tasks more efficiently than eixsting data-based (i.e., SEM-based) unsupervised RL methods (Liu & Abbeel, 2021b; Yarats et al., 2021).

## 2 PRELIMINARIES

### 2.1 MARKOV DECISION PROCESS (MDP)

We assume the environment modeled as an infinite-horizon[1] Markov Decision Process (MDP) $M = \langle S, A, T, r, \gamma, p_0 \rangle$, where $S$ is the set of states $s$, $A$ is the set of actions $a$, $T : S \times A \to \Delta(S)$ is the transition probability, $r : S \to \mathbb{R}$ is the reward function, $\gamma \in [0, 1]$ is the discount factor, $p_0 \in \Delta(S)$ is the initial state distribution. The policy $\pi : S \to \Delta(A)$ is a mapping from state distribution over actions[2]. For a given policy, its *state* stationary distribution $\bar{d}^\pi(s)$ and *state-action* stationary distribution $d^\pi(s, a)$ are defined as follows:

$$\bar{d}^\pi(s) := \begin{cases} (1 - \gamma) \sum_{t=0}^{\infty} \gamma^t \Pr(s_t = s) & \text{if } \gamma < 1, \\ \lim_{T \to \infty} \frac{1}{T+1} \sum_{t=0}^{T} \Pr(s_t = s) & \text{if } \gamma = 1, \end{cases} \quad (1)$$

$$d^\pi(s, a) := \begin{cases} (1 - \gamma) \sum_{t=0}^{\infty} \gamma^t \Pr(s_t = s, a_t = a) & \text{if } \gamma < 1, \\ \lim_{T \to \infty} \frac{1}{T+1} \sum_{t=0}^{T} \Pr(s_t = s, a_t = a) & \text{if } \gamma = 1, \end{cases} \quad (2)$$

where $s_0 \sim p_0$, $a_t \sim \pi(s)$, and $s_{t+1} \sim T(s_t, a_t)$ for each timestep $t \geq 0$. $d^\pi$ and $\bar{d}^\pi$ can be understood as normalized (discounted) occupancy measures of $(s, a)$ and $s$ respectively. For brevity, we focus on discounted MDPs ($\gamma < 1$) in the main text, but our formulation can be easily generalized to undiscounted ($\gamma = 1$) settings (see Appendix L). We will use the bar notation $(\bar{\cdot})$ to denote the distributions for state $s$, e.g., $\bar{d}^\pi(s)$. The goal of standard RL is to learn a policy that maximizes the expected rewards by interacting with the environment: $\max_\pi (1 - \gamma) \mathbb{E}_\pi \left[ \sum_{t=0}^{\infty} \gamma^t r(s_t) \right] = \mathbb{E}_{s \sim \bar{d}^\pi}[r(s)] = \langle \bar{d}^\pi, r \rangle$.

### 2.2 UNSUPERVISED RL VIA STATE ENTROPY MAXIMIZATION (SEM)

In the unsupervised pre-training of RL (Laskin et al., 2021), the agent is trained by interacting with a reward-free MDP $\langle S, A, T, \gamma, p_0 \rangle$ over long periods, where the goal is to learn a policy that can

---

[1]We consider infinite-horizon MDPs in the paper for simplicity, but our formulation can be extended to finite-horizon MDPs (Appendix M).

[2]We call $\pi$ Makovian policy if $\pi$ depends only on the last state and call it stationary policy if $\pi$ does not depend on timestep.

quickly adapt to downstream tasks defined by reward functions that are unknown a priori. In this work, we are particularly interested in the approach of maximizing the entropy of state stationary distribution (Jain et al., 2023; Mutti et al., 2021; 2022; Liu & Abbeel, 2021b; Yarats et al., 2021) as an objective for unsupervised pre-training RL:

$$\pi^* := \arg\max_\pi \mathbb{H}[\bar{d}^\pi(s)] = -\sum_s \bar{d}^\pi(s) \log \bar{d}^\pi(s) \tag{3}$$

In other words, we aim to pre-train a policy whose state visitations cover the entire state space equally well. Intuitively, having such an SEM policy would allow agents to efficiently receive reward signals even for arbitrarily sparse reward functions, facilitating fast adaptation for downstream tasks. More discussions on why learning an SEM policy can be useful for RL pre-training can be found in Appendix A.

## 2.3 Existing SEM Methods for RL Pre-training

Existing methods for SEM (Lee et al., 2020; Mutti et al., 2021; Liu & Abbeel, 2021b; Yarats et al., 2021) commonly follow the following procedures: (1) construct intrinsic reward functions $\hat{r}(s) \approx -\log \bar{d}^\pi(s)$, e.g., by particle-based state-entropy estimation: $\hat{r}(s_i) \approx -\log \bar{d}^\pi(s_i) \approx \log\left(\|s_i - s_i^{k\text{-NN}}\|_2\right)$ where $s_i^{k\text{-NN}}$ denotes $k$-nearest neighbor of $s_i$, and (2) update the policy parameters via policy gradients in the direction of maximizing $\hat{r}$. However, this procedure, in principle, requires *on-policy* state particles $\{s_i\}_{i=1}^N$ when estimating the state entropy of the current policy $\bar{d}^\pi(s)$, implying that the past experiences cannot be reused naively. Still, for off-policy learning to improve sample-efficiency, existing methods for SEM pre-training (Liu & Abbeel, 2021b; Yarats et al., 2021) naively use (off-policy) state particles in the replay buffer for entropy estimation to construct reward functions, combined with an off-policy RL algorithm. This may not be guaranteed to converge to an SEM policy as they estimate the replay buffer's state entropy, rather than the entropy of the target policy's state stationary distribution. Importance sampling can be adopted for off-policy corrections (Mutti et al., 2021) of policy gradients, but it suffers from high variance issue with the curse of horizon (Liu et al., 2018). Consequently, existing methods for SEM pre-training are either biased (off-policy) or sample-inefficient (on-policy). The key challenge in devising a sample-efficient SEM algorithm lies in estimating the entropy of the target policy's stationary state distribution using an arbitrary off-policy dataset.

## 3 SEMDICE

In this section, we derive our SEM method, *State-Entropy-Maximization via stationary DIstribution Correction Estimation (SEMDICE)*, which computes an SEM policy from arbitrary off-policy dataset. Our derivation starts by formulating the regularized SEM problem via concave programming that directly optimizes stationary distributions, rather than optimizing policy. All the proofs can be found in Appendix C.

### 3.1 Concave programming for SEM

Some careful readers may wonder which policy set should be considered to obtain an optimal SEM policy, as it may not be immediately evident that searching for an optimal SEM policy within stationary Markov policies (instead of richer non-Markovian policies) is sufficient. Fortunately, the following proposition addresses this concern.

**Proposition 3.1.** *(Hazan et al., 2019) There always exists a **stationary Markovian** policy that maximizes the entropy of state stationary distribution (i.e., solution of (3)). Such an optimal stationary Markovian policy is generally stochastic.*

By Proposition 3.1, it is sufficient to consider the stationary distribution induced by stationary Markovian policies to obtain an optimal SEM policy. We then begin our derivation with the following concave programming problem that optimizes the stationary distributions to solve a (regularized) SEM problem, where the primary objective is to maximize the entropy of state stationary

distribution of some target policy while adopting $f$-divergence regularization.

$$\max_{d, \bar{d} \geq 0} \underbrace{-\sum_s \bar{d}(s) \log \bar{d}(s)}_{\text{state entropy } \mathbb{H}[\bar{d}(s)]} \underbrace{-\alpha \mathrm{D}_f\left(d(s,a)||d^D(s,a)\right)}_{\text{concave regularizer}} \tag{4}$$

$$\text{s.t. } \sum_{a'} d(s',a') = (1-\gamma)p_0(s') + \gamma \sum_{s,a} d(s,a) T(s'|s,a) \ \ \forall s' \tag{5}$$

$$\sum_a d(s,a) = \bar{d}(s) \ \ \forall s \tag{6}$$

The Bellman flow constraint (5) guarantees that $d(s,a)$ is a valid state-action stationary distribution for some policy, where $d(s,a)$ can be understood as a normalized occupancy measure of $(s,a)$. The marginalization constraint (6) ensures that $\bar{d}(s)$ is the state stationary distribution directly induced by $d(s,a)$. In (4), $\mathrm{D}_f(d(s,a)||d^D(s,a)) := \mathbb{E}_{(s,a) \sim d^D}\left[f\left(\frac{d(s,a)}{d^D(s,a)}\right)\right]$ denotes the $f$-divergence between $d$ and $d^D$, $D = \{(s,a,s')_i\}_{i=1}^N$ denotes any off-policy dataset of interest (e.g., replay buffer of the agent), and $d^D$ is its corresponding distributions. We assume $f$ is a strictly convex function and continuously differentiable. For brevity, we will abuse the notation $d^D$ to represent $(s,a) \sim d^D$, $(s,a,s') \sim d^D$. The regularization hyperparameter $\alpha > 0$ balances between maximizing state entropy and preventing distribution shift from past experiences.

The regularization term $\mathrm{D}_f(d||d^D)$ in (4) can be understood as imposing trust-region updates Schulman et al. (2015) by constraining the solution to the vicinity of previous state-action visitations. Technically, the regularization term ensures strict concavity of the objective function with respect to $d$, thereby guaranteeing the uniqueness of the optimal solution for the concave programming (4-6). In contrast to existing DICE-based RL methods (Lee et al., 2021; 2022; Nachum & Dai, 2020) that formulate optimization problems only for *state-action* stationary distribution, we define the optimization problem that jointly optimizes *state* stationary distribution with the marginalization constraint (6) to deal with *state* entropy.

To sum up, we seek state-action stationary distribution $d$ (and its corresponding state stationary distribution $\bar{d}$) whose state entropy is being maximized. Once we have computed the optimal solution $(d^*, \bar{d}^*)$ of the concave programming (4-6), its corresponding optimal policy can be obtained by normalized $d^*$ for each state (Puterman, 1994): $\pi^*(a|s) = \frac{d^*(s,a)}{\bar{d}^*(s)}$.

## 3.2 LAGRANGE DUAL FORMULATION

Still, solving (4-6) directly requires a white-box model of the environment, which is inaccessible in many practical applications of RL. To derive an algorithm that can be fully optimized in a *model-free* manner, we consider the Lagrangian for the constrained optimization problem (4-6)

$$\max_{\bar{d},d \geq 0} \min_{\nu,\mu} -\sum_s \bar{d}(s) \log \bar{d}(s) - \alpha \mathrm{D}_f(d||d^D) + \sum_s \mu(s) \left(\sum_a d(s,a) - \bar{d}(s)\right) \tag{7}$$

$$+ \sum_{s'} \nu(s') \left((1-\gamma)p_0(s') + \gamma \sum_{s,a} d(s,a) T(s'|s,a) - \sum_{a'} d(s',a')\right)$$

where $\nu(s) \in \mathbb{R}$ are the Lagrange multipliers for the Bellman flow constraints (5), and $\mu(s) \in \mathbb{R}$ are the Lagrange multipliers for the marginalization constraints (6). Then, we rearrange the terms in (7):

$$\max_{\bar{d},d \geq 0} \min_{\nu,\mu} (1-\gamma)\mathbb{E}_{s_0 \sim p_0}[\nu(s_0)] - \alpha \mathbb{E}_{(s,a) \sim d^D}\left[f\left(\frac{d(s,a)}{d^D(s,a)}\right)\right] \tag{8}$$

$$+ \sum_{s,a} d(s,a) \left(\underbrace{\mu(s) + \gamma \mathbb{E}_{s'}[\nu(s')] - \nu(s)}_{=:e_{\nu,\mu}(s,a)}\right) - \sum_s \bar{d}(s)\left(\mu(s) + \log \bar{d}(s)\right)$$

$$= \min_{\nu,\mu} \max_{\bar{d},d \geq 0} (1-\gamma)\mathbb{E}_{s_0 \sim p_0}[\nu(s_0)] - \alpha \mathbb{E}_{d^D}\left[f\left(\frac{d(s,a)}{d^D(s,a)}\right)\right] + \sum_{s,a} d(s,a) e_{\nu,\mu}(s,a) - \sum_s \bar{d}(s)\left(\mu(s) + \log \bar{d}(s)\right) \tag{9}$$

In (9), we could reorder maximin to minimax thanks to strong duality (Boyd et al., 2004). Finally, using the Fenchel conjugate, we can eliminate the inner maximization problem and end up with a single convex minimization problem[3].

---

[3]This simplification from min-max to min was made possible by introducing the optimization variable $\bar{d}$ along with the marginalization constraint (6) (see Appendix B).

**Theorem 3.2.** *The minimax optimization problem (9) is equivalent to solving the following uncon-strained minimization problem:*

$$\min_{\nu,\mu}(1-\gamma)\mathbb{E}_{p_0}[\nu(s_0)] + \mathbb{E}_{d^D}\left[\alpha f_+^*\left(\tfrac{1}{\alpha}e_{\nu,\mu}(s,a)\right)\right] + \sum_s \exp(-\mu(s)-1) =: L(\nu,\mu) \qquad (10)$$

*where $f_+^*(y) = \max_{x\geq 0} xy - f(x)$. The objective function $L(\nu,\mu)$ is convex for $\nu$ and $\mu$. Fur-thermore, given the optimal solution $(\nu^*,\mu^*) = \arg\min_{\nu,\mu} L(\nu,\mu)$, the stationary distribution corrections of the optimal SEM policy are given by:*

$$\frac{d^*(s,a)}{d^D(s,a)} = (f')^{-1}\left(\frac{1}{\alpha}\left(\underbrace{\mu^*(s)+\gamma\mathbb{E}_{s'}[\nu^*(s')]-\nu^*(s)}_{=e_{\nu^*,\mu^*}(s,a)}\right)\right)_+ =: w_{\nu^*,\mu^*}^*(s,a) \qquad (11)$$

*where $x_+ := \max(0,x)$.*

In short, by operating in the space of stationary distributions, computing a stationary Markovian SEM policy can, in principle, be addressed by solving a *convex minimization* problem. The resulting policy can be obtained by $\pi^*(a|s) = \frac{d^*(s,a)}{\sum_{a'} d^*(s,a')} = \frac{w_{\nu^*,\mu^*}^*(s,a)d^D(s,a)}{\sum_{a'} w_{\nu^*,\mu^*}^*(s,a')d^D(s,a')}$ in the case of finite MDPs.

### 3.3 PRACTICAL ALGORITHM

Still, one practical issue in (10) is that optimizing it is unstable due to its inclusion of $\exp(\cdot)$ term, often causing exploding gradient problems. To remedy this issue, we consider the follow-ing numerically stable alternative objective function $\widetilde{L}$ that replaces $\sum_s \exp(-\mu(s)-1)$ with $\log\sum_s \exp(-\mu(s))$.

**Theorem 3.3.** *Define the objective functions $\widetilde{L}(\nu,\mu)$ as*

$$\widetilde{L}(\nu,\mu) := (1-\gamma)\mathbb{E}_{s_0}[\nu(s_0)] + \mathbb{E}_{(s,a)\sim d^D}\left[\alpha f_+^*\left(\tfrac{1}{\alpha}e_{\nu,\mu}(s,a)\right)\right] + \textcolor{red}{\log\sum_s \exp(-\mu(s))} \qquad (12)$$

*Then, for any optimal solutions $(\nu^*,\mu^*) = \arg\min_{\nu,\mu} L(\nu,\mu)$ and $(\widetilde{\nu}^*,\widetilde{\mu}^*) = \arg\min_{\nu,\mu}\widetilde{L}(\nu,\mu)$, the following holds:*

$$L(\nu^*,\mu^*) = \widetilde{L}(\widetilde{\nu}^*,\widetilde{\mu}^*) \qquad (13)$$

*Also, there exists a constant $C$ such that the following holds:*

$$\mu^* = \widetilde{\mu}^* + C \quad \text{and} \quad \nu^* = \widetilde{\nu}^* + \tfrac{C}{1-\gamma} \qquad (14)$$

Note that the gradient $\nabla_s \sum_s \log\sum_s \exp(-\mu(s)) = -\frac{\exp(-\mu(s))}{\sum_{s'} \exp(-\mu(s'))}\nabla_s\mu(s)$ normalizes $\exp(\cdot)$ by softmax, thus $\widetilde{L}$ no longer suffer from numerical instability by large gradients. At first glance, it seems that optimizing (12) could yield a solution that is completely different from that of (10). However, Theorem 3.3 shows that their optimal objective function values are the same and their optimal solutions only differ in a constant shift. Furthermore, despite its constant shift, it does not change anything for $w^*$ computation in (11), as can be seen as follows:

$$e_{\widetilde{\nu}^*,\widetilde{\mu}^*}(s,a) = \left(\mu^*(s)-C\right)+\gamma\left(\mathbb{E}[\nu^*(s')]-\tfrac{C}{1-\gamma}\right)-\left(\nu^*(s)-\tfrac{C}{1-\gamma}\right)$$

$$= \mu^*(s)+\gamma\mathbb{E}[\nu^*(s')]-\nu^*(s) = e_{\nu^*,\mu^*}(s,a)$$

$$\therefore w_{\nu^*,\mu^*}^*(s,a) = w_{\widetilde{\nu}^*,\widetilde{\mu}^*}^*(s,a)$$

Still, (12) requires summing up (or integrating) every possible state, which is intractable for large (or continuous) state space. Therefore, we approximate the $\log\sum\exp(-\mu(s))$ term via Monte-Carlo integration with an arbitrary distribution $q$: $\log\sum\exp(-\mu(s)) = \log\mathbb{E}_q[\exp(-\mu(s)-\log q(s))]$. Finally, we optimize the following objective function with $q(s) = \bar{d}^D(s)$:

$$\min_{\nu,\mu}(1-\gamma)\mathbb{E}_{s_0}[\nu(s_0)] + \mathbb{E}_{(s,a)\sim d^D}\left[\alpha f_+^*\left(\tfrac{1}{\alpha}e_{\nu,\mu}(s,a)\right)\right] \qquad (15)$$

$$+ \log\mathbb{E}_{s\sim\bar{d}^D(s)}\left[\exp(-\mu(s)-\log\bar{d}^D(s))\right]$$

The last remaining challenge is that (15) still cannot be naively optimized in a fully model-free manner in continuous domains, as it contains computing expectations w.r.t. the transition model inside the non-linear function $f_+^*(\cdot)$. For a practical implementation, we use the following objective that can be easily optimized via sampling only from $d^D$:

$$\min_{\nu,\mu}(1-\gamma)\mathbb{E}_{p_0}[\nu(s_0)] + \mathbb{E}_{(s,a,s')\sim d^D}\left[\alpha f_+^*\left(\tfrac{1}{\alpha}\hat{e}_{\nu,\mu}(s,a,s')\right)\right] \tag{16}$$

$$+ \log\mathbb{E}_{s\sim\bar{d}^D(s)}\left[\exp(-\mu(s)-\log\bar{d}^D(s))\right] =: \hat{L}(\nu,\mu)$$

where $\hat{e}(s,a,s') := \mu(s) + \gamma\nu(s') - \nu(s)$ is a single-sample estimate of $e(s,a)$. Note that every term in (16) can now be evaluated only using the $(s,a,s')$ samples from $D$, thus we can optimize $\nu$ and $\mu$ easily. Although (16) is a biased estimate of (15), one can easily show that $\hat{L}(\nu,\mu)$ is an upper bound of $L(\nu,\mu)$, i.e. $L(\nu,\mu) \leq \hat{L}(\nu,\mu)$ always holds, where the equality holds when the MDP is deterministic, by Jensen's inequality and the convexity of $f_+^*(\cdot)$. Once we get the optimal solution $(\hat{\nu}^*,\hat{\mu}^*) = \arg\min_{\nu,\mu}\hat{L}(\nu,\mu)$, we end up with the stationary distribution corrections of the optimal SEM policy: $\frac{d^{\pi^*}(s,a)}{d^D(s,a)} = w_{\hat{\nu}^*,\hat{\mu}^*}(s,a)$ by (11). In practice, to deal with continuous state space, we parameterize $\nu_\theta : S \to \mathbb{R}$ and $\mu_\omega : S \to \mathbb{R}$ using simple MLPs, which take the state $s$ as an input and output a scalar value, and optimize the network parameters.

**Policy Extraction**  The final remaining question is how to obtain an explicit policy, as our method computes the distribution correction ratios $w^*$ in (11) instead of the policy itself directly. We follow the i-projection used in (Lee et al., 2021).

$$\min_\pi \mathbb{KL}\left(d^D(s)\pi(a|s)||d^D(s)\pi^*(a|s)\right) \tag{17}$$

$$= \mathbb{E}_{\substack{s\sim d^D \\ a\sim\pi}} - [\log w^*(s,a) - \mathrm{D_{KL}}(\pi(\bar{a}|s)||\pi_D(\bar{a}|s))] + C \tag{18}$$

Essentially, minimizing (18) corresponds to computing a policy $\pi(\cdot|s)$ that mimics the optimal SEM policy $\pi^*(\cdot|s)$ for each state $s \in D$. In summary, SEMDICE computes the SEM policy as follows: First, minimize (16), and Second, extract the policy using the obtained $w_{\nu^*,\mu^*}$ by i-projection (18). In practice, rather than training until convergence at each iteration, we perform a single gradient update for $\nu$, $\mu$, and $\pi$. We outline the details of policy extraction (Appendix F) and the full learning procedure with pseudo-code in Algorithm 1 (Appendix G).

*Remark* 3.4. Minimizing $\widetilde{L}(\nu,\mu)$ in (15) is equivalent to solving the original (regularized) state-entropy-maximization problem (4-6), demonstrating that computing a (regularized) SEM policy can, in principle, be achieved by arbitrary off-policy dataset $D$. Note that this approach directly optimizes the state entropy of a target policy, rather than optimizing the state entropy of the replay buffer as in (Yarats et al., 2021; Liu & Abbeel, 2021b). To the best of our knowledge, SEMDICE is the first principled and practical off-policy SEM algorithm that can learn from arbitrary off-policy experiences. Also, (Hazan et al., 2019) showed that state entropy is not concave for policy parameters, whereas state entropy is concave for the stationary distribution space $\bar{d}(s)$. That being said, our method, which directly optimizes stationary distributions, may offer better convergence properties compared to existing policy-based algorithms. However, providing a formal analysis for the convergence guarantee remains as future work.

# 4  RELATED WORK

**Unsupervised RL and State Entropy Maximization**  Our work falls within the realm of unsupervised reinforcement learning, specifically addressing the challenge of task-agnostic exploration. The reward-free exploration framework has gained significant attention in recent years (Jin et al., 2020; Tarbouriech et al., 2020; Kaufmann et al., 2020). While these methods share a similar context to our work, they pursue largely orthogonal objectives.

State entropy maximization is a particular instance of reward-free exploration objective, where the agent aims to estimate the density of states and maximize entropy (Hazan et al., 2019; Lee et al., 2020; Liu & Abbeel, 2021b; Yarats et al., 2021; Mutti et al., 2022; Tiapkin et al., 2023; Kim et al., 2023; Yang & Spaan, 2023). However, these methods mostly rely on policy gradients with the

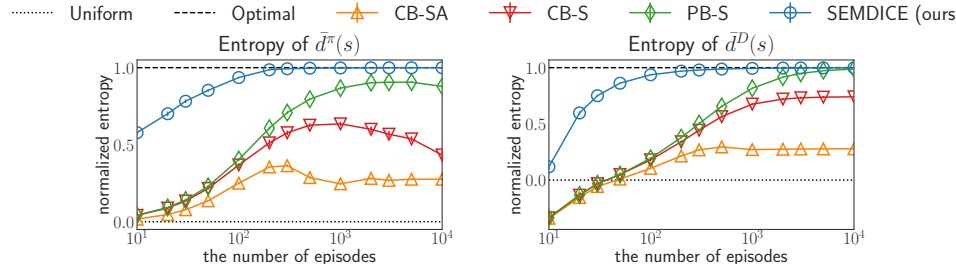

Figure 1: Performance of SEMDICE and baselines in randomly generated tabular MDPs. The first column indicates normalized policy entropy (0: state entropy of uniform policy, 1: state entropy of optimal SEM policy), the second column indicates normalized entropy of the cumulative experiences. We observe that only SEMDICE converges to an optimal SEM policy, whereas baselines do not directly maximize the state entropy.

reward function for the estimated state entropy, and they are either sample inefficient due to their requirements of on-policy samples (Hazan et al., 2019; Mutti et al., 2022), or optimize the biased objective due to estimating the state entropy of the replay buffer (Liu & Abbeel, 2021b; Yarats et al., 2021). In contrast, our method operates directly in the space of stationary distributions and serves as the first principled off-policy SEM method.

Besides state entropy maximization, various self-supervised objectives have been explored for unsupervised RL pre-training (Laskin et al., 2021). The goal of pre-training in this context is to compute a policy without relying on task-specific reward functions, enabling rapid adaptation to future, unknown downstream tasks defined by reward functions. Unsupervised RL pre-training algorithms are categorized in three main ways (Laskin et al., 2021). **Knowledge-based** methods aim to increase knowledge about the environment by maximizing prediction error (Pathak et al., 2017; Burda et al., 2019; Pathak et al., 2019). **Competence-based** methods (Lee et al., 2020; Eysenbach et al., 2019; Liu & Abbeel, 2021a; Laskin et al., 2022) aim to learn explicit skill representations by maximizing the mutual information between encoded observation and skill. Lastly, **data-based** methods (Liu & Abbeel, 2021b; Yarats et al., 2021) aim to achieve data diversity via particle-based entropy maximization. Our SEMDICE is categorized into the data-based method when considering unsupervised RL pre-training.

**Stationary DIstribution Correction Estimation (DICE)** DICE-family algorithms perform stationary distribution estimation and have shown significant promise in various off-policy learning scenarios in reinforcement learning (RL), such as off-policy evaluation (Nachum et al., 2019a; Zhang* et al., 2020; Zhang et al., 2020; Yang et al., 2020), reinforcement learning (Lee et al., 2021; Kim et al., 2024; Mao et al., 2024), constrained RL (Lee et al., 2022), imitation learning (Kim et al., 2022b; Ma et al., 2022; Kim et al., 2022a; Sikchi et al., 2024), and more. However, to the best of our knowledge, no DICE-based algorithms have been proposed to address state entropy maximization.

## 5 EXPERIMENTS

In this section, we empirically evaluate our SEMDICE and baselines: to demonstrate (1) SEMDICE converge to an optimal SEM policy, (2) visualization of SEMDICE's state visitation, (3) how efficiently SEMDICE pre-trained policy can adapt to downstream tasks on URL benchmarks (Laskin et al., 2021). Ablation experiments on the choice of $f$ and $\alpha$ for SEMDICE can be found in the Appendix K.

### 5.1 STATE ENTROPY MAXIMIZATION IN FINITE MDPS

We first evaluate SEMDICE and baseline algorithms on randomly generated finite MDPs with 20 states and 4 actions to show how effectively SEMDICE maximizes the state entropy, compared to baselines, when optimized using the off-policy dataset (the entire replay buffer). We conduct repeated experiments with different seeds for 100 runs.

**Baselines** Baselines are policy-gradient-based methods, with different (non-stationary) intrinsic reward functions $\hat{r}$. We consider the following baselines, and they can be thought of as analogies to existing unsupervised RL approaches. (1) **CB-SA**: it constructs the state-action-count-based exploration rewards by $\hat{r}(s,a) = \frac{1}{\sqrt{N(s,a)}}$, where $N(s,a)$ is the cumulative $(s,a)$-visitation counts by the agent. This baseline can be understood as an analogy to the existing count-based methods or RND (Burda et al., 2019). (2) **CB-S**: it constructs state-count-based rewards $\hat{r}(s) = \frac{1}{\sqrt{N(s)}}$, where $N(s)$ is the $s$-visitation counts. It is similar to CB-SA but only considers state visitation frequencies, thus it is expected to be closer to state-entropy maximization. (3) **PB-S**: it constructs entropy-based rewards $\hat{r}(s) = -\log d^D(s)$, where $d^D$ is the empirical state distribution of the cumulative experiences (which is **not** direct estimate of $-\log d^\pi(s)$). This baseline can be understood as an analogy to existing SEM methods performing off-policy updates (Liu & Abbeel, 2021b; Yarats et al., 2021). (4) **Uniform**: sample actions uniformly at random.

For all baselines, we first collect trajectories using the current policies and then construct the non-stationary rewards function based on visitation counts/data-state-entropy measures accordingly. Lastly, we perform policy gradient ascents along the direction of maximizing the constructed reward function.

**Evalution** For each run, we (1) generate a random MDP and initialize the policy as uniform policy; (2) For each method, use the current policy $\pi$ to collect 10 episodes and add them to the replay buffer $D$; (3) Using $D$, compute the MLE transition matrix $\hat{T}$, the intrinsic reward function $\hat{r}$, and the value function $Q_{\hat{r}}^\pi$ with respect to the intrinsic reward; (4) Perform off-policy policy gradient ascent based on the estimated $Q_{\hat{r}}^\pi$; (5) Repeat steps (2)-(4) until $D$ contains 1000 episodes. Throughout the process, we compute the state entropy of the current policy $\bar{d}^\pi(s)$ as well as the entropy of the replay buffer $\bar{d}^D(s)$.

**Results** Figure 1 presents the results. We observe that **CB-S** baseline generates a slightly better SEM policy than **CB-SA**, which is natural as it focuses on the exploration of the unvisited states while ignoring action uncertainty. **PB-S** baseline could obtain nearly maximal state-entropy for the *dataset* in the dataset $\bar{d}^D(s)$, but it is not directly optimizing the target policy's state entropy $\bar{d}^\pi(s)$. **SEMDICE** is the only algorithm that could converge to an optimal SEM policy efficiently. It also achieves much better sample efficiency in terms of the state entropy of dataset that **PB-S**. Although we didn't include the result, we also tested *on-policy* policy-gradient approach for SEM. It could converge to an optimal SEM policy with a carefully chosen learning rate, but it required $\times 100$ more samples to reach near-optimal entropy maximization. This results confirm that SEMDICE is capable of computing an optimal SEM policy even from the arbitrary off-policy dataset. In Appendix I, we further show that SEMDICE still converges to the optimal SEM policy even when the dataset is collected by a purely off-policy agent (i.e. uniform random policy).

## 5.2 STATE ENTROPY MAXIMIZATION: VISUALIZATION

This section aims to visualize the behavior of SEMDICE and baselines for their state visitations during policy learning. To this end, we use MountainCar (Continuous) from Gymnasium (Towers et al., 2023), a classic control task with a low-dimensional continuous state and action space. The task involves a deterministic MDP where a car must escape from a sinusoidal valley by building momentum. The observation space is two-dimensional, consisting of `position` $\in [-1.2, 0.6]$ and `velocity` $\in [-0.07, 0.07]$, while actions correspond to applying directional forces in the range $[-1, 1]$. The low dimensionality allows for clear visualization of state visitation distributions in the 2D plane, facilitating straightforward comparisons between different agents.

**Baselines** We compare SEMDICE to baselines from three categories of unsupervised RL. **Knowledge-based** methods include RND (Burda et al., 2019), which relies on prediction error as an exploration signal. **Data-based** methods aim to directly maximize state entropy, which can be our direct interest, and include ProtoRL (Yarats et al., 2021), APT (Liu & Abbeel, 2021b), and MEPOL (Mutti et al., 2021). **Competence-based** methods such as DIAYN (Eysenbach et al., 2019) and CIC (Laskin et al., 2022) focus on learning diverse skills by maximizing the mutual information between skills and observations.

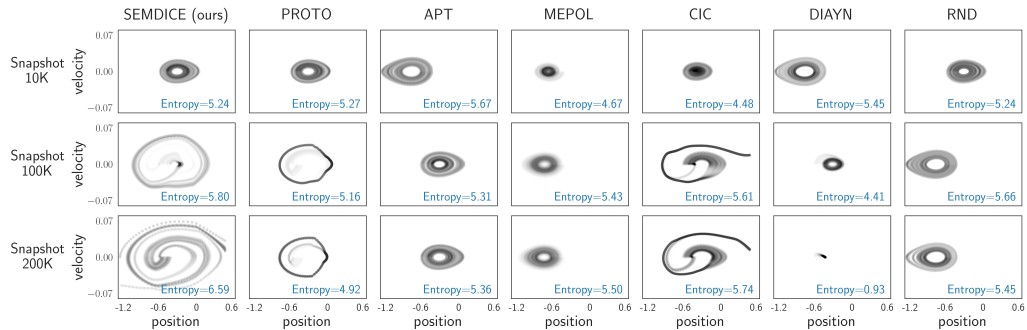

Figure 2: Mountaincar state coverage: Visualization of agent pretraining results across SEMDICE and baselines. During the reward-free pretraining stage, we save model snapshots at 10K, 100K, and 200K steps. From each snapshot, we draw 30k samples via environment interactions for visualizations. The state space is discretized into uniform $51 \times 51$ bins, and the empirical distribution is normalized to compute the entropy shown at the bottom of each subplot. While existing methods failed to converge to a state-entropy-maximizing policy, SEMDICE successfully did so.

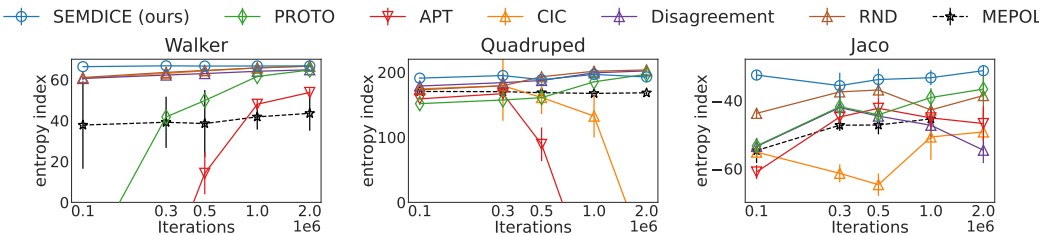

Figure 3: Particle-based entropy estimations of pretrained SEMDICE and baselines in URLB (Laskin et al., 2021). We follow the setup from URLB and pretrain each agent with 2M environment interactions. We save policy snapshots at $[0.1M, 0.3M, 0.5M, 1M, 2M]$ steps, and report the change of entropy index along the training process. We observe that SEMDICE achieves maximum state entropy in both Walker and Jaco Arm domain, and close to maximum state entropy in Quadruped domain. It is also the most sample efficient approach amongst all agents: with as few as $100K$ environment interactions, SEMDICE learns a proper state entropy maximizing policy. For example, in Walker domain, multiple agents like RND, Disagreement, and PROTO managed to achieve similar state entropy as SEMDICE. However, they are much less sample efficient, and only converged to maximum state entropy policy after close to $2M$ environment interactions.

**Evaluation and Results** We evaluate all methods based on their overall state coverage and entropy during the reward-free pretraining phase. Specifically, we take snapshots of each policy at 10K, 100K, and 200K environment steps under purely intrinsic rewards. For each snapshot, we collect 30,000 state samples by interacting with the environment using the fixed policy, and visualize the corresponding state visitation distribution. To compute the entropy of the stationary state distribution induced by each policy, we discretize the 2D continuous observation space into a $51 \times 51$ grid and count the number of visits to each grid cell. Note that all methods operate directly in continuous state and action spaces; discretization is used solely for entropy computation.

Figure 2 shows that existing state entropy maximization methods (PROTO, APT, MEPOL) fail to converge to a single SEM policy. Instead, their learned policies exhibit non-stationary behavior throughout training, likely due to their dependence on a non-stationary, learned reward function. In contrast, SEMDICE progressively increases state coverage over time and eventually converges to a single, stable SEM policy, which is in contrast to the oscillatory behaviors observed in the baselines.

## 5.3 URL BENCHMARK

Finally, we evaluate the state-entropy maximization and fine-tuning RL performance using tasks from URLB (Laskin et al., 2021). URLB consists of twelve tasks across three different domains: Walker, Quadruped, and Jaco Arm. During the pre-training phase, SEMDICE and baselines com-

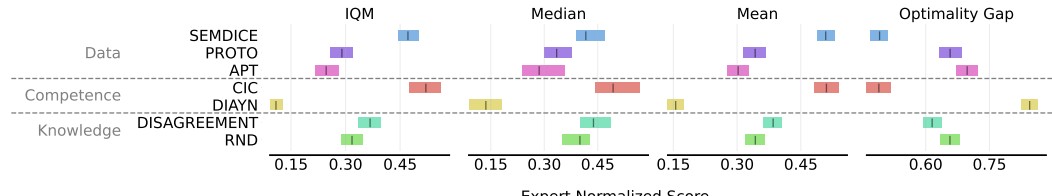

Figure 4: We evaluate all methods on the 12 tasks provided by the URLB benchmark (Laskin et al., 2021), covering three domains: Walker (stand, walk, flip, run) and Quadruped (stand, walk, run, jump) are locomotion tasks that require maintaining balance and strategically controlling the body; Jaco Arm (reach bottom left, bottom right, top left, top right) consists of manipulation tasks involving precise control of a 6-DOF robotic arm with a three-finger gripper to reach specified target positions. To ensure statistically reliable comparisons, we use `rliable` (Agarwal et al., 2021) to report aggregate statistics across multiple seeds and tasks, along with stratified bootstrap confidence intervals. Each plot is generated using a total of 120 runs (3 domains × 4 tasks/domain × 10 seeds).

putes the pre-trained policy without task-specific reward, solely relying on their intrinsic objective functions. Once the policy pre-training is done after 2M steps, a small amount of environment interactions are allowed with the task-specific rewards (100K steps). We then evaluate how efficiently each method adapts to the given downstream tasks, specifically examining how well the pre-trained policies serve as a good initialization for rapid adaptation. We compared SEMDICE with the baselines provided by URLB. We follow the experimental protocoal of URLB: we ran 10 seeds per task, and used the same hyperparameters for baselines, except for using the smaller MLP hidden size $1024 \rightarrow 256$ for fast training/evaluation.

**Results** In Figure 3, we report the state entropy of the policy checkpoints during pre-training per method, and SEMDICE shows the highest state entropy estimates across all domains. In Figure 4, we present the fine-tuning performance. Overall, SEMDICE significantly outperforms all data-based (i.e. SEM-based) and knowledge-based baselines in terms of rapid adaptation performance during fine-tuning phase. This highlights that state entropy maximization is a desirable objective for RL pre-training, as illustrated in Appendix A. Although SEMDICE underperforms CIC, a competence-based URL method, we believe its performance can be further enhanced by integrating representation learning techniques (e.g., those used in CIC). It is important to emphasize that our results were achieved solely through the SEM objective without additional mechanisms. Moreover, SEMDICE's potentially more unbiased and efficient SEM optimization could have resulted in good policy initialization for fine-tuning. Qualitatively, we observed that even the data-based baselines sometimes exhibit static behavior, while SEMDICE's learned policy consistently showed actively moving behavior across different domains.

## 6 CONCLUSION

We presented SEMDICE, a new state-entropy-maximization (SEMDICE) algorithm for RL pre-training. Existing SEM methods rely on policy gradients, and they either require on-policy samples or perform (off-policy) biased optimization due to its estimation of state entropy of replay buffer. In contrast, SEMDICE directly optimizes in the space of stationary distributions, and our derivation shows that computing an optimal SEM policy can be achieved by solving a single convex minimization problem with arbitrary off-policy dataset. To the bast of our knowledge, SEMDICE is the first principled off-policy SEM algorithm. Through various tabular and continuous domain experiments, we demonstrated that SEMDICE converges to a single stationary Markov SEM policy, and outperforms baselines in terms of maximizing state entropy. As for future work, we plan to incorporate representation learning components into SEMDICE so that it enables faster meaningful feature detection and better navigation in high-dimensional domains such as pixel-based domains. Another promising future direction is to devise a DICE-based method for competence-based unsupervised RL algorithms (e.g. maximizing the mutual information), which can serve as a principled off-policy algorithm for it.

## ACKNOWLEDGMENTS

This work was supported by NSF AI4OPT AI Centre. Pieter Abbeel holds concurrent appointments as a Professor at UC Berkeley and as an Amazon Scholar. This paper describes work performed at UC Berkeley and is not associated with Amazon. This work was partly supported by Institute of Information & communications Technology Planning & Evaluation (IITP) grant funded by the Korea government (MSIT) (No. RS-2024-00457882, AI Research Hub Project) and the Institute of Information & Communications Technology Planning & Evaluation (IITP) grant (RS-2020-II201361, Artificial Intelligence Graduate School Program (Yonsei University).

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

## A   WHY STATE-ENTROPY MAXIMIZATION FOR RL PRE-TRAINING

In this section, we illustrate why state-entropy maximization can be a good choice for RL pre-training. Consider the following information-theoretic *regularized regret objective* defined by (Eysenbach et al., 2022).

$$\text{ADAPTATIONOBJECTIVE}\big(\bar{d}_{\text{pre}}(s), r(s)\big) := \min_{\bar{d}^*(s)} \max_{\bar{d}^+(s)} \underbrace{\mathbb{E}_{\bar{d}^+(s)}[r(s)] - \mathbb{E}_{\bar{d}^*(s)}[r(s)]}_{\text{regret}} + \text{D}_{\text{KL}}(\bar{d}^*(s)||\bar{d}_{\text{pre}}(s))$$

(19)

In the ADAPTATIONOBJECTIVE, for a given pre-trained policy's stationary distribution $\bar{d}_{\text{pre}}$ and a downstream reward function $r$, it measures the regret of the policy ($\bar{d}^*$) after adaptation from the pre-trained policy ($\bar{d}_{\text{pre}}$) with the information cost, where optimal $\bar{d}^+(s)$ denotes the state stationary distribution of the optimal policy for the reward $r$ (introduced to define regret). Then, a desirable pre-trained policy (or its corresponding $\bar{d}^\pi$) can be defined in terms of minimizing the (regularized) regret against the *worst-case* reward function:

$$\min_{\bar{d}_{\text{pre}}(s)} \max_{r(s)} \text{ADAPTATIONOBJECTIVE}\big(\bar{d}_{\text{pre}}(s), r(s)\big)$$

(20)

(Eysenbach et al., 2022) has shown that (20) is equivalent to:

$$\min_{\bar{d}_{\text{pre}}(s)} \max_{\bar{d}^+(s)} \text{D}_{\text{KL}}(\bar{d}^+(s)||\bar{d}_{\text{pre}}(s)),$$

(21)

and thus maximum-entropy (uniform) $\bar{d}_{\text{pre}}$ is the solution of (21) and (20). This means that SEM policy provides robust policy initialization against the worst-case reward function for fine-tuning, justifying why state-entropy maximization is useful for unsupervised RL pre-training. Figure 5 visualizes information geometry of RL pre-training (Eysenbach et al., 2022).

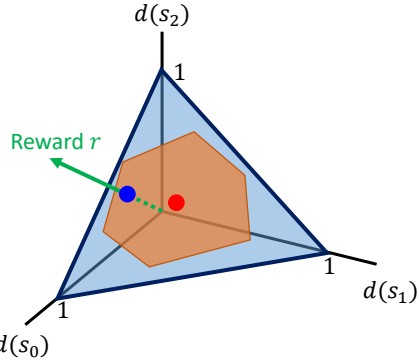

Figure 5: Visualization of reward function and policies as their state stationary distributions in a 3-state MDP. The shaded orange area denotes a set of achievable state stationary distributions that lie on the probability simplex $\Delta(S)$ (shaded blue). The green arrow represents the reward function $r$ as a vector starting at the origin. ● denotes the state stationary distribution of a pretrained SEM policy, $\bar{d}_{\text{pre}} = [\frac{1}{3}, \frac{1}{3}, \frac{1}{3}]$, and ● denotes the state stationary distribution of an optimal policy for $r$, $\bar{d}^+$ which is an intersection of $r$ and $\Delta(S)$. The distance from ● to ● reflects the adaptation cost.

## B    WHY BOTH $d(s,a)$ AND $\bar{d}(s)$ ARE USED

Without $\bar{d}(s)$, the main objective becomes:

$$-\sum_{s,a} d(s,a) \log \sum_{a'} d(s,a') \tag{22}$$

This makes algorithm derivation complicated due to the summation inside logarithm. Specifically, our derivation with $\bar{d}(s)$ results in the following inner-max problem (See (23) in Appendix C):

$$\max_{w\geq 0} \mathbb{E}_{d^D}\Big[w(s,a)\big(\mu(s)+\gamma\mathbb{E}_{s'}[\nu(s')]-\nu(s)\big)-\alpha f\big(w(s,a)\big)\Big] = \alpha f_+^*\big(\tfrac{1}{\alpha}\big(\mu(s)+\gamma\mathbb{E}_{s'}[\nu(s')]-\nu(s)\big)$$

where inner-maximization problem $\max_{w\geq 0}$ could have been easily eliminated by applying Fenchel conjugate of $f$, i.e. $f_+^*(y) := \max_{x\geq 0} xy - f(x)$.

In contrast, without $\bar{d}(s)$, we have the following inner-maximization problem:

$$\max_{w\geq 0} \mathbb{E}_{(s,a)\sim d^D}\Big[w(s,a)\Big(-\log \sum_{a'} w(s,a')d^D(s,a') + \gamma\mathbb{E}_{s'}[\nu(s')]-\nu(s)\Big)-\alpha f\big(w(s,a)\big)\Big]$$

where Fenchel conjugate is not directly applicable to eliminate this inner-maximization problem.

To sum up, the introduction of the new optimization variable $\bar{d}(s)$ eliminates the need for solving nested min-max optimization, enabling SEM to be formulated as solving a single convex optimization problem.

## C    PROOFS

### C.1    PROOF OF PROPOSITION 3.1

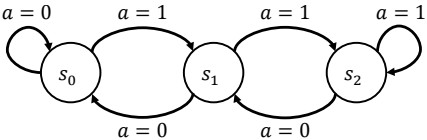

Figure 6: An illustrative example of MDP with three states and two actions, where the optimal SEM policy is stochastic. Specifically, in order to maximize the state entropy, the action selection should be randomized at $s_1$. If the agent deterministically takes action $a=0$ at $s_1$, it precludes visiting $s_2$. Conversely, if the agent deterministically takes action $a=1$ at $s_1$, it will never visit $s_0$ thereafter.

**Proposition 3.1.** *(Hazan et al., 2019) There always exists a **stationary Markovian** policy that maximizes the entropy of state stationary distribution (i.e., solution of (3)). Such an optimal stationary Markovian policy is generally stochastic.*

**Lemma C.1.** *(Puterman, 1994) For any possibly non-Markovian policy $\pi$, define a stationary Markov policy $\pi'$ as $\pi'(a|s) = \frac{d_\pi(s,a)}{d_\pi(s,a)}$. Then, $d_\pi = d_{\pi'}$.*

*Proof.* Firstly, it is evident that there must exist at least one non-stationary, non-Markovian policy capable of maximizing the entropy of the state's stationary distribution. Then, according to Lemma C.1, we can always obtain its corresponding stationary and Markov policy. This derived policy is generally stochastic, as any deterministic policy can be arbitrarily bad in terms of maximization of state entropy, as exemplified in Figure 6. □

### C.2    PROOF OF THEOREM 3.2

**Theorem 3.2.** *The minimax optimization problem (9) is equivalent to solving the following unconstrained minimization problem:*

$$\min_{\nu,\mu}(1-\gamma)\mathbb{E}_{p_0}[\nu(s_0)] + \mathbb{E}_{d^D}\Big[\alpha f_+^*\big(\tfrac{1}{\alpha}e_{\nu,\mu}(s,a)\big)\Big] + \sum_s \exp(-\mu(s)-1) =: L(\nu,\mu) \tag{10}$$

where $f_+^*(y) = \max_{x \geq 0} xy - f(x)$. *The objective function $L(\nu, \mu)$ is convex for $\nu$ and $\mu$. Furthermore, given the optimal solution $(\nu^*, \mu^*) = \arg\min_{\nu,\mu} L(\nu, \mu)$, the stationary distribution corrections of the optimal SEM policy are given by:*

$$\frac{d^*(s, a)}{d^D(s, a)} = (f')^{-1}\left(\tfrac{1}{\alpha}\left(\underbrace{\mu^*(s) + \gamma\mathbb{E}_{s'}[\nu^*(s')] - \nu^*(s)}_{=e_{\nu^*,\mu^*}(s,a)}\right)\right)_+ =: w_{\nu^*,\mu^*}^*(s, a) \qquad (11)$$

*where $x_+ := \max(0, x)$.*

*Proof.* Start with Equation (9)

$$\min_{\nu,\mu} \max_{\bar{d}, d \geq 0} (1 - \gamma)\mathbb{E}_{s_0 \sim p_0}[\nu(s_0)] - \alpha\mathbb{E}_{d^D}\left[f\left(\underbrace{\tfrac{d(s,a)}{d^D(s,a)}}_{=:w(s,a)}\right)\right] + \sum_{s,a} d(s,a)e_{\nu,\mu}(s,a) - \sum_s \bar{d}(s)\left(\mu(s) + \log\bar{d}(s)\right)$$

$$= \min_{\nu,\mu} \max_{\bar{d}, w \geq 0} (1 - \gamma)\mathbb{E}_{s_0}[\nu(s_0)] - \alpha\mathbb{E}_{d^D}\left[f\left(w(s,a)\right)\right] + \sum_{s,a} d^D(s,a)w(s,a)e_{\nu,\mu}(s,a) - \sum_s \bar{d}(s)\left(\mu(s) + \log\bar{d}(s)\right)$$

$$(23)$$

First, we can derive the closed-form solution for $\bar{d}$ in (23):

$$\frac{\partial L}{\partial\bar{d}(s)} = -(\mu(s) + \log\bar{d}(s)) - 1 = 0 \qquad (24)$$

$$\Rightarrow \bar{d}^*(s) = \exp(-\mu(s) - 1) \qquad (25)$$

Check the second order conditions to confirm that $L$ is concave with respect to $\bar{d}(s)$:

$$\frac{\partial^2 L}{\partial\bar{d}(s)^2} = -\frac{1}{\bar{d}(s)} < 0 \qquad (26)$$

We can plug this solution to $L$, which eliminates the $\max_{\bar{d}}$:

$$\min_{\nu,\mu} \max_{w \geq 0} (1 - \gamma)\mathbb{E}_{s_0}[\nu(s_0)] - \alpha\mathbb{E}_{d^D}\left[f\left(w(s,a)\right) - \tfrac{1}{\alpha}w(s,a)e_{\nu,\mu}(s,a)\right] + \sum_s \exp(-\mu(s) - 1)$$

$$(27)$$

Consider simplifying the second term in (27) using Fenchel conjugate:

$$\max_{w \geq 0} -\alpha\mathbb{E}_{d^D}\left[f\left(w(s,a)\right) - \tfrac{1}{\alpha}w(s,a)e_{\nu,\mu}(s,a)\right] \qquad (28)$$

$$\max_{w \geq 0} \alpha\mathbb{E}_{d^D}\left[\tfrac{1}{\alpha}w(s,a)e_{\nu,\mu}(s,a) - f\left(w(s,a)\right)\right] \qquad (29)$$

$$= \alpha\mathbb{E}_{d^D}\left[\max_{w(s,a)\geq 0} w(s,a)\left(\tfrac{1}{\alpha}e_{\nu,\mu}(s,a)\right) - f\left(w(s,a)\right)\right] \qquad (30)$$

$$= \alpha\mathbb{E}_{d^D}\left[f_+^*\left(\tfrac{1}{\alpha}e_{\nu,\mu}(s,a)\right)\right] \qquad (31)$$

Hence, the Equation (9) is equivalent to:

$$\min_{\nu,\mu} (1 - \gamma)\mathbb{E}_{p_0}[\nu(s_0)] + \mathbb{E}_{d^D}\left[\alpha f_+^*\left(\tfrac{1}{\alpha}e_{\nu,\mu}(s,a)\right)\right] + \sum_s \exp(-\mu(s) - 1) =: L(\nu, \mu)$$

Consider obtaining the closed-form solution for the inner-maximization for $w$ in (27):

$$\frac{\partial L}{\partial w(s,a)} = -\alpha d^D(s,a)f'\left[w(s,a)\right] + d^D(s,a)e_{\nu,\mu}(s,a) = 0 \qquad (32)$$

$$\Rightarrow w^*(s,a) = f'^{-1}\left(\tfrac{1}{\alpha}e_{\nu,\mu}(s,a)\right)_+ \qquad (33)$$

where $x_+ := \max(0, x)$. $\qquad\qquad\square$

Again, we plug $w^*(s, a)$ into the original objective function, which results in the minimization problem:

$$\min_{\nu, \mu}(1 - \gamma)\mathbb{E}_{s_0}\big[\nu(s_0)\big] - \alpha\mathbb{E}_{d^D}\Big[f\Big(f'^{-1}\Big(\tfrac{e_{\nu,\mu}(s,a)}{\alpha}\Big)_+\Big) - \tfrac{1}{\alpha}f'^{-1}\Big(\tfrac{e_{\nu,\mu}(s,a)}{\alpha}\Big)_+ e_{\nu,\mu}(s, a)\Big]$$
$$+ \sum_s \exp(-\mu(s) - 1)$$

We can check that the function $L$ is convex with respect to both $\nu$ and $\mu$, by noting that $f_+^*(\cdot)$ is a convex function.

### C.3  PROOF OF THEOREM 3.3

**Theorem 3.3.** *Define the objective functions $\widetilde{L}(\nu, \mu)$ as*

$$\widetilde{L}(\nu, \mu) := (1 - \gamma)\mathbb{E}_{s_0}\big[\nu(s_0)\big] + \mathbb{E}_{(s,a)\sim d^D}\Big[\alpha f_+^*\Big(\tfrac{1}{\alpha}e_{\nu,\mu}(s, a)\Big)\Big] + \log\sum_s \exp(-\mu(s)) \tag{12}$$

*Then, for any optimal solutions $(\nu^*, \mu^*) = \arg\min_{\nu,\mu} L(\nu, \mu)$ and $(\widetilde{\nu}^*, \widetilde{\mu}^*) = \arg\min_{\nu,\mu} \widetilde{L}(\nu, \mu)$, the following holds:*

$$L(\nu^*, \mu^*) = \widetilde{L}(\widetilde{\nu}^*, \widetilde{\mu}^*) \tag{13}$$

*Also, there exists a constant $C$ such that the following holds:*

$$\mu^* = \widetilde{\mu}^* + C \quad\text{and}\quad \nu^* = \widetilde{\nu}^* + \tfrac{C}{1-\gamma} \tag{14}$$

**Lemma C.2.** *For any functions $\nu$ and $\mu$, and a constant $C$, the following equality holds:*

$$\widetilde{L}(\nu, \mu) = \widetilde{L}(\nu + \tfrac{C}{1-\gamma}, \mu + C). \tag{34}$$

$$\widetilde{L}(\nu + \tfrac{C}{1-\gamma}, \mu + C) \tag{35}$$

$$= (1 - \gamma)\mathbb{E}_{s_0\sim p_0}[\nu(s_0) + \tfrac{C}{1-\gamma}] + \log\sum \exp(-\mu(s) - C) \tag{36}$$

$$\quad + \mathbb{E}_{(s,a)\sim d^D}\Big[\alpha f_+^*\Big(\tfrac{1}{\alpha}\big(\mu(s) + \cancel{C} + \gamma\mathbb{E}_{s'}[\nu(s')] + \tfrac{\gamma\cancel{C}}{\cancel{1-\gamma}} - \nu(s) - \tfrac{\cancel{C}}{\cancel{1-\gamma}}\big)\Big)\Big]$$

$$= (1 - \gamma)\mathbb{E}_{s_0\sim p_0}[\nu(s_0)] + \cancel{C} + \mathbb{E}_{(s,a)\sim d^D}\Big[\alpha f_+^*\Big(\tfrac{1}{\alpha}e_{\nu,\mu}(s, a)\Big)\Big] + \log\sum \exp(-\mu(s)) - \cancel{C} \tag{37}$$

$$= (1 - \gamma)\mathbb{E}_{s_0\sim p_0}[\nu(s_0)] + \mathbb{E}_{(s,a)\sim d^D}\Big[\alpha f_+^*\Big(\tfrac{1}{\alpha}e_{\nu,\mu}(s, a)\Big)\Big] + \log\sum \exp(-\mu(s)) \tag{38}$$

$$= \widetilde{L}(\nu, \mu) \tag{39}$$

**Lemma C.3.** *For any functions $\nu$ and $\mu$, $L(\nu, \mu) \geq \widetilde{L}(\nu, \mu)$ holds. The equality holds if and only if*

$$\sum_s \exp(-\mu(s) - 1) = 1. \tag{40}$$

*Proof.* For any $x \geq 0$, the inequality $x - 1 \geq \log x$ always holds, where the equality holds if and only if $x = 1$. By applying this equality, we have:

$$\sum_s \exp(-\mu(s) - 1) - 1 \geq \log\sum_s \exp(-\mu(s) - 1) \tag{41}$$

$$\Leftrightarrow \sum_s \exp(-\mu(s) - 1) \cancel{-1} \geq \log\sum_s \exp(-\mu(s)) \cancel{\exp(-1)} \tag{42}$$

$$\Leftrightarrow \sum_s \exp(-\mu(s) - 1) \geq \log\sum_s \exp(-\mu(s)) \tag{43}$$

$$\Leftrightarrow L(\nu, \mu) \geq \widetilde{L}(\nu, \mu) \tag{44}$$

where the equality holds if and only if $\sum_s \exp(-\mu(s) - 1) = 1$. $\qquad\square$

**Lemma C.4.** *Let $(\widetilde{\nu}^*, \widetilde{\mu}^*) = \arg\min_{\nu,\mu} \widetilde{L}(\nu, \mu)$. Then, there exists a constant $C$ such that $(\widetilde{\nu}^* + C, \widetilde{\mu}^* + \frac{C}{1-\gamma})$ is an optimal solution of $\min_{\nu,\mu} L(\nu, \mu)$.*

*Proof.* Let $C^* = \log \sum_s \exp(-\widetilde{\mu}^*(s) - 1)$ be a constant. Also, let $\bar{\nu}^* = \widetilde{\nu}^* + \frac{C^*}{1-\gamma}$ and $\bar{\mu}^* = \widetilde{\mu}^* + C^*$. Then,

$$\sum_s \exp\left(\bar{\mu}^*(s) - 1\right) = \sum_s \exp\left(-(\widetilde{\mu}^*(s) + C^*) - 1\right)$$

$$= \sum_s \exp\left(-\widetilde{\mu}^*(s) - 1 - \log\sum_{s'} \exp(-\widetilde{\mu}^*(s') - 1)\right)$$

$$= \frac{\sum_s \exp(-\widetilde{\mu}^*(s) - 1)}{\sum_{s'} \exp(-\widetilde{\mu}^*(s') - 1)} = 1. \tag{45}$$

Then, we have

$$L(\bar{\nu}^*, \bar{\mu}^*) = \widetilde{L}(\bar{\nu}^*, \bar{\mu}^*) \qquad \text{(by (45) and Lemma C.3)} \tag{46}$$

$$= \widetilde{L}\left(\widetilde{\nu}^*, \widetilde{\mu}^*\right) \qquad \text{(by Lemma C.2)} \tag{47}$$

$$= \min_{\nu,\mu} \widetilde{L}(\nu, \mu) \tag{48}$$

$$\leq \min_{\nu,\mu} L(\nu, \mu) \qquad \text{(by Lemma C.3)} \tag{49}$$

which implies that $(\bar{\nu}^*, \bar{\mu}^*)$ is the optimal solution of $\min_{\nu,\mu} L(\nu, \mu)$. $\qquad\square$

## D   F-DIVERGENCE

In the experiments, we use the softened version of chi-square divergence for $f$:

$$f_{\text{soft-}\chi^2}(x) := \begin{cases} x\log x - x + 1 & \text{if } 0 < x < 1 \\ \frac{1}{2}(x-1)^2 & \text{if } x \geq 1. \end{cases} \quad \Rightarrow \quad (f_{\text{soft-}\chi^2}(x)')^{-1}(x) = \begin{cases} \exp(x) & \text{if } x < 0 \\ x + 1 & \text{if } x \geq 0 \end{cases}$$

When $f = f_{\text{soft-}\chi^2}$, its corresponding $f_+^*(y) = \max_{x\geq 0} xy - f(x)$ is given by:

$$f_+(x) = \begin{cases} \infty & \text{if } x \leq 0 \\ x\log x - x + 1 & \text{if } 0 < x < 1 \\ \frac{1}{2}(x-1)^2 & \text{if } x \geq 1 \end{cases} \quad \Rightarrow \quad f_+^*(y) = \begin{cases} \exp(y) - 1 & \text{if } y < 0 \\ \frac{1}{2}y^2 + y & \text{if } y \geq 0 \end{cases} \tag{50}$$

where $f_+^*$ is the Fenchel conjugate of $f_+$.

## E   HYPERPARAMETERS

Baseline hyperparameters are taken from URLB (Laskin et al., 2021), except for using the smaller hidden sizes $1000 \rightarrow 256$ for fast training/evaluation.

| SEMDICE pretraining hyper-parameter | Value |
|---|---|
| Action repeat | 1 |
| $\alpha$ | 0.5 |
| Batch size | 1024 |
| $f$-type | softchiq |
| Hidden dimension | 256 |
| Learning rate | 0.0001 |
| nsteps | 1 |
| Update every step | 2 |

Table 1: Hyperparameter Settings

For the policy network, we use a tanh-Gaussian distributions, following the baselines in URLB.

## F   POLICY EXTRACTION

Our practical SEMDICE optimizes (16) that yields $\nu^*$ and $\mu^*$. However, $\nu^*$ itself is not a directly executable policy, so we should extract a policy from them. Note that the optimal SEM policy $\pi^*$ is encoded in $w^*$:

$$\frac{d^*(s,a)}{d^D(s,a)} = (f')^{-1}\Big(\frac{1}{\alpha}\Big(\underbrace{\mu^*(s) + \gamma\mathbb{E}_{s'}[\nu^*(s')] - \nu^*(s)}_{=e_{\nu^*,\mu^*}(s,a)}\Big)\Big)_+ =: w^*(s,a) \tag{11}$$

Then, we extract a policy from $w^*$ via I-projection policy extraction method (Lee et al., 2021).

$$\min_\pi \; \mathbb{KL}\big(d^D(s)\pi(a|s)||d^D(s)\pi^*(a|s)\big) \tag{51}$$

$$= \mathbb{E}_{\substack{s\sim d^D\\a\sim\pi}} - [\log w^*(s,a) - \mathrm{D_{KL}}(\pi(\bar{a}|s)||\pi_D(\bar{a}|s))] + C \tag{52}$$

$$= \mathbb{E}_{\substack{s\sim d^D\\a\sim\pi}} - \Big[\log(f')^{-1}\Big(\frac{1}{\alpha}\Big(e_{\nu^*,\mu^*}(s,a)\Big)\Big)_+ \cancel{-\mathrm{D_{KL}}(\pi(\bar{a}|s)||\pi_D(\bar{a}|s))}\Big] + C \tag{53}$$

$$\approx \mathbb{E}_{\substack{s\sim d^D\\a\sim\pi}} - \Big[\log(f')^{-1}\Big(\frac{1}{\alpha}e(s,a)\Big)_+\Big] + C \tag{54}$$

where $C$ is some constraint and $\pi^D(a|s)$ is a data policy. As we perform online optimization, $\pi_D$ can be considered as an old policy, and we ignored the KL term in our practical implementation. Finally, to enable $e_{\nu^*,\mu^*}$ to be evaluated every action $a$, we train an additional parametric function (implemented as an MLP that takes $(s,a)$ as an input and outputs a scalar value) by minimizing the mean squared error:

$$\min_e \mathbb{E}_{(s,a,s')\sim d^D}\Big[\big(e(s,a) - \hat{e}_{\nu^*,\mu^*}(s,a,s')\big)^2\Big] \tag{55}$$

In the following section, we present the pseudo-code for practical SEMDICE.

## G PSEUDO-CODE FOR SEMDICE

To sum up, SEMDICE computes a SEM policy by optimizing $(\nu^*, \mu^*)$, which corresponds to obtaining a stationary distribution correction ratios of the optimal SEM policy. Then, we extract a policy by training a $e$-network and performing I-projection as described in the previous section.

We assume $\nu, \mu, e, \pi$ are parameterized by $\theta, \omega, \psi, \phi$ respectively. Then, we optimize the parameters via stochastic gradient descent. The loss functions are summarized in the following:

$$J(\nu_\theta, \mu_\omega) := (1-\gamma)\mathbb{E}_{s_0}\left[\nu_\theta(s_0)\right] + \mathbb{E}_{(s,a,s')\sim d^D}\left[\alpha f_+^*\left(\frac{1}{\alpha}\hat{e}_{\nu_\theta,\mu_\omega}(s,a,s')\right)\right]$$
$$+ \log\mathbb{E}_{s\sim\bar{d}^D(s)}\left[\exp(-\mu_\omega(s) - \log\bar{d}^D(s))\right] \tag{56}$$

$$J(e_\psi) := \mathbb{E}_{(s,a,s')\sim d^D}\left[\left(e_\psi(s,a) - \hat{e}_{\nu_\theta,\mu_\omega}(s,a,s')\right)^2\right] \quad \text{(by (55))} \tag{57}$$

$$J(\pi_\phi) := \mathbb{E}_{\substack{s\sim d^D \\ a\sim\pi_\phi}} - \left[\log(f')^{-1}\left(\frac{1}{\alpha}e_\psi(s,a)\right)_+\right] \quad \text{(by 54)} \tag{58}$$

where $\hat{e}_{\nu_\theta,\mu_\omega}(s,a,s') = \mu_\omega(s) + \gamma\nu_\theta(s') - \nu_\theta(s)$. For $-\log\bar{d}^D(s)$ in (56), we used particle-based density estimation using $k$-nearest-neighbor: $-\log\bar{d}^D(s_i) \approx \log\left(\|s_i - s_i^{k\text{-NN}}\|_2\right)$.[4] Instead of fully optimizing $\nu_\theta$ and $\mu_\omega$ until convergence, we alternatively perform single gradient updates for $\nu_\theta$, $\mu_\omega$, $e_\psi$, and $\pi_\phi$. The pseudocode of SEMDICE is presented in Algorithm 1.

---

**Algorithm 1** SEMDICE

**Input:** Neural networks $\nu_\theta$, $\mu_\omega$, and $e_\psi$ with parameters $\theta$, $\omega$, and $\psi$, policy network $\pi_\phi$ with parameter $\phi$, replay buffer $D$, a learning rate $\eta$, a regularization hyperparameter $\alpha$

1: **for** each timestep $t$ **do**
2:     Sample an action $a_t \sim \pi_\phi(s_t)$ for the current state $s_t$.
3:     Observe next state $s_{t+1} \sim P(\cdot|s_t, a_t)$ by taking action to the environment.
4:     Add transition to replay buffer $D \leftarrow D \cup (s_t, a_t, s_{t+1})$
5:     Sample a minibatch from $D$ for the following SGD updates.
6:     Perform SGD updates:

$$(\theta, \omega) \leftarrow (\theta, \omega) - \eta\nabla_{\theta,\omega}J(\nu_\theta, \mu_\omega) \ (Eq.\ (56))$$
$$\psi \leftarrow \psi - \eta\nabla_\psi J(e_\psi) \ (Eq.\ (57))$$
$$\phi \leftarrow \phi - \eta\nabla_\phi J(\pi_\phi) \ (Eq.\ (58))$$

7: **end for**

---

[4]Though our practical also uses (approximate) estimation of $-\log d^D(s)$ via kNN, it does not contradict the main motivation of the paper. In (56), we use Monte Carlo integration to approximate $\log\sum\exp(-\mu(s))$ as $\log\hat{\mathbb{E}}_{d^D}[\exp(-\mu(s) - \log d^D(s))]$. While this approximation may introduce bias, it still aims to maximize the entropy of the target policy's state stationary distribution from an arbitrary off-policy dataset, rather than maximizing the state entropy of the replay buffer.

In contrast, existing methods focus on maximizing the intrinsic reward $\hat{r}(s) \approx -\log d^D(s)$, which corresponds to maximizing the state entropy of the replay buffer. That being said, they are approximating $-\log d^\pi(s)$ by $-\log d^D(s)$. As a result, existing methods optimize the biased objective even when the kNN approximation error vanishes, whereas SEMDICE optimizes the unbiased objective once the kNN approximation error vanishes. See also Appendix I, which shows that SEMDICE can compute the optimal SEM policy even with a dataset collected by a uniform random policy. This behavior of computing an optimal SEM policy from random datasets cannot be achieved by other baselines.

## H    ADDITIONAL EXPERIMENTS ON TABULAR MDPS - COMPARISON WITH MAXENT (HAZAN ET AL., 2019)

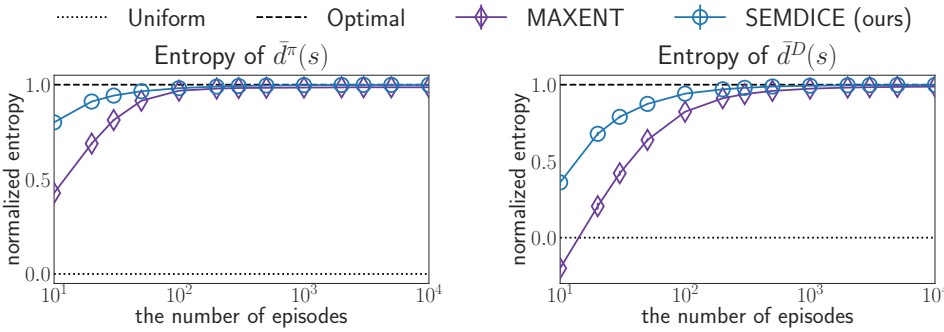

Figure 7: Performance of SEMDICE and MaxEnt (Hazan et al., 2019) in randomly generated tabular MDPs. The first column indicates normalized policy entropy (0: state entropy of uniform policy, 1: state entropy of optimal SEM policy), the second column indicates normalized entropy of the cumulative experiences.

We did not compare SEMDICE with MaxEnt (Hazan et al., 2019) in the main text because Max-Ent is not designed to compute a single policy but rather to obtain a set of policies. Specifically, it maximizes the entropy of the state distribution resulting from the weighted average of state distributions visited by multiple policies. This approach does not align with the scenario we consider, which focuses on fine-tuning a single policy derived from a pre-trained policy. Still, for quantitative comparison with MaxEnt, we have conducted additional experiments in tabular MDPs. In the result, MaxEnt (with tuned hyperparameters) underperforms SEMDICE in terms of learning efficiency. We also would like to emphasize that SEMDICE computes only a single stationary Markov policy, whereas MaxEnt continuously increases the number of policies it maintains, making it computationally inefficient.

## I    ADDITIONAL EXPERIMENTS ON TABULAR MDPS - SEMDICE USING DATASET COLLECTED BY UNIFORM POLICY

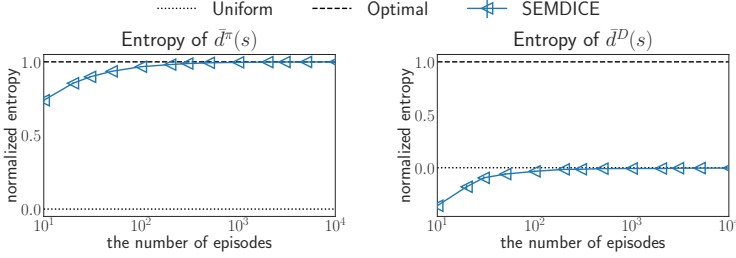

Figure 8: Performance of SEMDICE when the dataset is always being collected by uniform random policy in randomly generated tabular MDPs. The first column indicates normalized policy entropy (0: state entropy of uniform policy, 1: state entropy of optimal SEM policy), and the second column indicates normalized entropy of the cumulative experiences. This result demonstrates the capability of SEMDICE that can be optimized from arbitrary off-policy experiences.

## J   ADDITIONAL EXPERIMENTS ON TABULAR MDPS: VALUE-BASED BASELINES

In Figure 1, we presented the performance of the policy-based baselines: for each iteration, they compute policy gradients for the intrinsic reward $\hat{r}$ and perform policy gradient ascents. In this section, we provide an additional result for value-based RL baselines. The intrinsic rewards are defined in the same way as described in the main text.

$$\textbf{CB-SA} : \hat{r}(s,a) = \frac{1}{\sqrt{N(s,a)}}, \quad \textbf{CB-S} : \hat{r}(s,a) = \frac{1}{\sqrt{N(s)}}, \quad \textbf{PB-S} : \hat{r}(s,a) = -\log d^D(s) \quad (59)$$

where $N(s,a)$ is the cumulative $(s,a)$-visitation counts by the agent, $N(s)$ is the $s$-visitation counts by the agent, and $d^D(s)$ is the empirical state distribution of the cumulative experiences. Then, for each value-based baselines, they maintain $Q$-table (initialized by zeros) and perform Q-learning updates by:

$$Q(s,a) \leftarrow Q(s,a) + \eta\Big(\hat{r}(s,a) + \gamma \max_{a'} Q(s',a') - Q(s,a)\Big) \quad (60)$$

where $\eta \in (0,1)$ is a learning rate.

As value-based RL methods do not maintain explicit policy, we evaluated two types of target policy induced by $Q$: greedy (deterministic) policy ($\pi(s) = \arg\max_a Q(s,a)$) and softmax policy ($\pi(a|s) \propto \exp(Q(s,a)/\tau)$). For exploration, all methods adopt an $\epsilon$-soft behavior policy with $\epsilon \propto O(1/T)$ decreasing over time. We also decrease the temperature of softmax policy over time with the rate of $O(1/T)$. The result for greedy policy is shown in Figure 9, and the result for the softmax policy is presented in Figure 10.

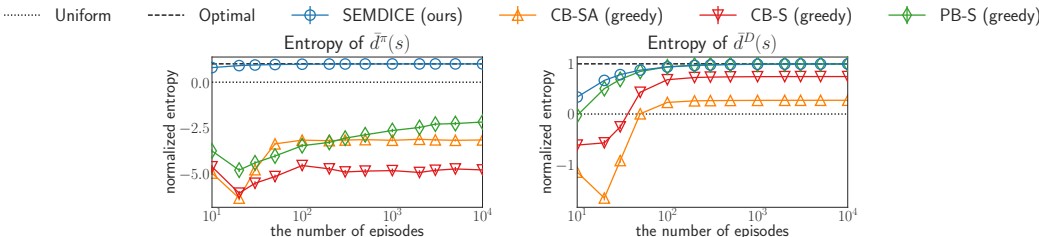

Figure 9: Performance of SEMDICE and value-based baselines in randomly generated tabular MDPs, where the target policy $\pi(s)$ of baseline is given by the **greedy** policy w.r.t. $Q(s,a)$.

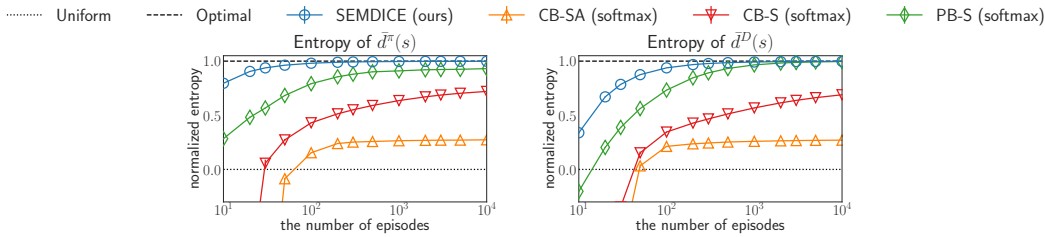

Figure 10: Performance of SEMDICE and value-based baselines in randomly generated tabular MDPs, where the target policy $\pi(a|s)$ of baseline is given by the **softmax** policy w.r.t. $Q(s,a)$.

As can be seen from the left column of Figure 9, the greedy target policy for the estimated $Q$ exhibits very low state entropy. This result is expected, as the optimal SEM policy should generally be *stochastic* (Proposition 3.1), but the greedy target policy is *deterministic*. Thus, it does NOT converge to the optimal stochastic SEM policy, leading to significant suboptimality. The entropy of the cumulative state-visit experiences $\mathbb{H}[\bar{d}^D(s)]$ (the right column of Figure 9) for PB-S approaches to the maximum state entropy, but it was achieved by mixture of many non-stationary deterministic policies. In contarst, SEMDICE yields a single stationary stochastic SEM policy.

In Figure 10, we can see that adopting the softmax (thus stochastic) policy leads to better state entropy for the target policy (left column). However, there is no guarantee that any of these value-based baselines with the softmax policy will converge to an optimal SEM policy.

# K    ABLATION EXPERIMENT RESULTS FOR SEMDICE

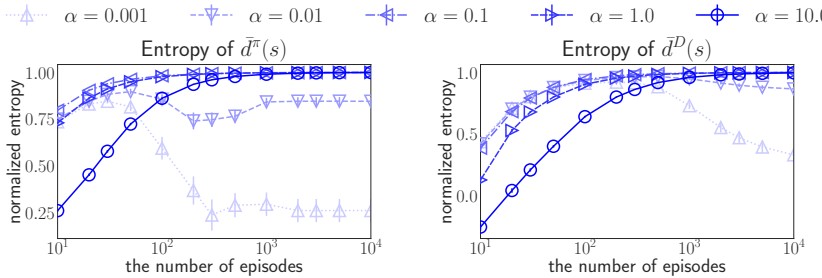

Figure 11: Experimental results on varying $\alpha$ (SEMDICE without function approximation).

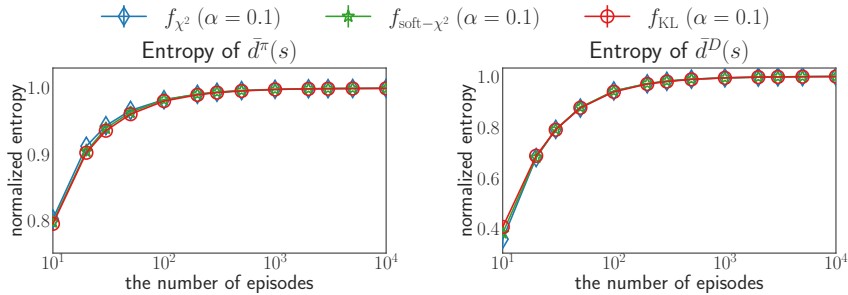

Figure 12: Experimental results on different $f$-divergence (SEMDICE without function approximation).

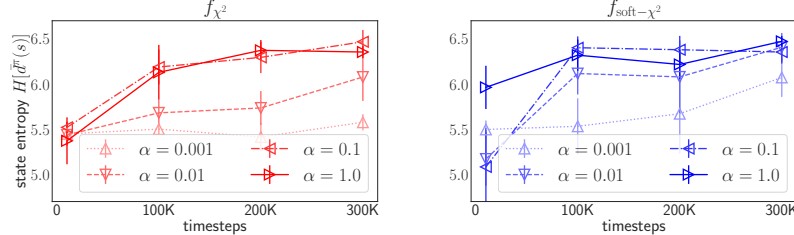

Figure 13: The effect of the varying $\alpha$ and $f$ on ContinuousMountainCar (SEMDICE with neural function approximation).

As shown in Figure 11-13 SEMDICE can become numerically unstable with very small values of $\alpha$, but it remains not too sensitive to $\alpha$ as long as it is within a reasonable range. Additionally, using different $f$ didn't make significant difference.

## L   GENERALIZATION TO UNDISCOUNTED ($\gamma = 1$) SETTING

SEMDICE can be generalized to deal with $\gamma = 1$ setting, where $\bar{d}^\pi(s) := \lim_{T \to \infty} \frac{1}{T+1} \sum_{t=0}^T \Pr(s_t = s; \pi)$. To this end, the original optimization problem (3-5) in the paper should be modified by adding an additional normalization constraint $\sum_{s,a} d(s,a) = 1$. Then, for any $\gamma \in (0, 1]$,

$$\max_{d, \bar{d} \geq 0} - \sum_s \bar{d}(s) \log \bar{d}(s) - \alpha \mathrm{D}_f \left( d(s,a) || d^D(s,a) \right) \tag{61}$$

$$\text{s.t. } \sum_{a'} d(s', a') = (1 - \gamma) p_0(s') + \gamma \sum_{s,a} d(s,a) T(s'|s,a) \ \ \forall s' \tag{62}$$

$$\sum_a d(s,a) = \bar{d}(s) \ \ \forall s \tag{63}$$

$$\textcolor{red}{\sum_{s,a} d(s,a) = 1} \tag{64}$$

By taking the similar derivation steps of SEMDICE with the additional constraint (64), we can show that solving the following unconstrained convex optimization problem is equivalent to solving (61-64), which additionally includes a single scalar variable $\lambda$:

$$\min_{\nu, \mu, \textcolor{red}{\lambda}} (1 - \gamma) \mathbb{E}_{p_0}[\nu(s_0)] + \mathbb{E}_{d^D} \left[ \alpha f_+^* \left( \tfrac{1}{\alpha} e_{\nu, \mu, \lambda}(s,a) \right) \right] + \sum_s \exp(-\mu(s) - 1) \textcolor{red}{- \lambda} \tag{65}$$

where $e_{\nu, \mu, \lambda}(s,a) := \lambda + \mu(s) + \gamma \mathbb{E}_{s'}[\nu(s')] - \nu(s)$ and $\lambda$ is Lagrange multiplier for the normalization constraint (64). The detailed derivation will be included in the final version of the paper.

## M   EXTENSION TO UNDISCOUNTED (NON-STATIONARY) FINITE-HORIZON SETTING

SEMDICE can be extended to a finite horizon setting, where the goal is to maximize the entropy of the average state distribution $\bar{d}^\pi(s) := \frac{1}{T+1} \sum_{t=0}^T \Pr(s_t = s; \pi)$. The following formulation results in a non-stationary (timestep-dependent) policy by $\pi_t(a|s) = \frac{d_t^*(s,a)}{\sum_{a'} d_t^*(s,a')}$.

$$\max_{d_t, \bar{d}_t \geq 0} - \sum_{t=0}^T \sum_s \bar{d}_t(s) \log \bar{d}_t(s) - \alpha \sum_{t=0}^T \mathrm{D}_f \left( d_t(s,a) || d^D(s,a) \right) \tag{66}$$

$$\text{s.t. } \sum_a d_0(s,a) = p_0(s) \tag{67}$$

$$\forall s \quad \sum_{a'} d_t(s', a') = \sum_{s,a} d_{t-1}(s,a) T(s'|s,a) \ \ \forall s', t \in \{1, \ldots, T\} \tag{68}$$

$$\sum_a d_t(s,a) = \bar{d}_t(s) \ \ \forall s, t \in \{0, \ldots, t\} \tag{69}$$

We can show that solving (66-69) is equivalent to solving the following unconstrained convex optimization problem, where the difference to the original objective function is that $\nu, \mu$ are timestep-dependent (i.e. $\nu_t, \mu_t$).

$$\min_{\{\nu_t\}_{t=0}^T, \{\mu_t\}_{t=0}^T} (1 - \gamma) \mathbb{E}_{p_0}[\nu_0(s_0)] + \textcolor{red}{\sum_{t=0}^T} \mathbb{E}_{d^D} \left[ \alpha f_+^* \left( \tfrac{1}{\alpha} e_{t, \nu, \mu}(s,a) \right) \right] + \textcolor{red}{\sum_{t=0}^T} \sum_s \exp(-\mu_t(s) - 1) \tag{70}$$

where $e_{t, \nu, \mu}(s,a) := \mu_t(s) + \mathbb{E}_{s'}[\nu_{t+1}(s')] - \nu_t(s)$, and $\nu_{T+1}(\cdot) := 0$. The detailed derivation will be included in the final version of the paper.

## N   MORE DISCUSSIONS ON SEMDICE'S OBJECTIVE

Despite its inclusion of $f$-divergence regularization, SEMDICE is not fundamentally opposed to particle-based state-entropy estimation; rather, it addresses the same objective of maximizing state entropy, with the additional inclusion of $f$-divergence regularization to enhance the stability of the learning process. It is important to note that this regularizer is not intended to impose a strong constraint on the optimal policy, and its influence can be controlled by adjusting the hyperparameter $\alpha$. Note that many successful standard reward-maximizing RL algorithms have made use of regularizations such as entropy (Haarnoja et al., 2018) and KL-divergence (Schulman et al., 2015), f-divergence (Nachum et al., 2019b), Bregman-divergence (Geist et al., 2019), and so on. Although these regularizations might appear contrary to pure reward maximization, their inclusion actually stabilizes the overall learning process, and improves the overall reward performance of the resulting policy. In a similar vein, our state-entropy maximization method employs f-divergence regularization to prevent abrupt distribution shift (analogous to TRPO), enhancing the robustness and stability of learning process. Additionally, from a technical perspective, f-divergence regularization makes the objective function strictly concave, guaranteeing the global optimality of any local optimum (i.e. ensuring the uniqueness of the optimal solution).

Also, SEMDICE's objective function indeed aims to maximize the state entropy, i.e. encouraging exploration of unvisited states more. To see this more intuitively, consider the main objective function of OptiDICE (Lee et al., 2021), a reward-maximizing RL algorithm:

$$\min_{\nu}(1-\gamma)\mathbb{E}_{s_0}\big[\nu(s_0)\big] + \mathbb{E}_{(s,a,s')\sim d^D}\Big[\alpha f_+^*\Big(\tfrac{1}{\alpha}\big(r(s,a)+\gamma\nu(s')-\nu(s)\big)\Big)\Big] \tag{71}$$

Then, the following is the main objective function of our SEMDICE, a state-entropy maximization method:

$$\min_{\nu,\mu}(1-\gamma)\mathbb{E}_{s_0}\big[\nu(s_0)\big] + \mathbb{E}_{(s,a,s')\sim d^D}\Big[\alpha f_+^*\Big(\tfrac{1}{\alpha}\big(\mu(s)+\gamma\nu(s')-\nu(s)\big)\Big)\Big] + \log\sum_s\exp(-\mu(s)) \tag{72}$$

That being said, the for a fixed $\mu$, SEMDICE can be interpreted as OptiDICE with the reward function defined by $\mu(s)$. Of course, we are jointly optimizing the $\mu$ instead of using fixed $\mu$. Also, due to the second term of the objective function ($f_+^*$ is increasing function), $\mu(s)$ is pressured to decrease more in regions of high data density and less where the data density is low. Consequently, SEMDICE's resulting policy will tend to explore unvisited states more actively, which is well-aligned with the goal of state entropy maximization.

## O   MACHINE AND SETUP

We run the experiments on machines with Titan XP GPUs. Pretraining took approximately 10 hrs and finetuning took around 30 minutes.

