# OpenReview forum: "SEMDICE: Off-policy State Entropy Maximization via Stationary Distribution Correction Estimation"
_ICLR.cc/2025/Conference — ICLR 2025 Poster_

### Official Review · Reviewer_qDon · 2024-10-24

**Soundness:** 3
**Presentation:** 3
**Contribution:** 3
**Rating:** 6
**Confidence:** 2

**Summary:**

This paper addresses the issue of state entropy maximization by off-policy and principled method.
Key idea underlying this work is the Eq.4-6, where the authors jointly optimize state stationary distribution and state-action stationary distribution.
The method is evaluated in both toy examples and tasks in URLB.

**Strengths:**

1, There is a theoretical analysis (Lagrange dual formulation and convex problem simplification) for the proposed approach. While I have not checked the proofs carefully, they appear to be correct.

2, The goal of the paper for addressing the challenge of state entropy maximization in RL is well motivated.

3, The results in the experiments clearly support the claims made in the paper.

**Weaknesses:**

1, Why the performance (entropy index) of APT and CIC drop after training in Fig.3 in Quadruped experiment?

2, Will the code be publicly available after acceptance? Some details of the specific implementation are not given.

3, Line 296: inimizing -> minimizing

**Questions:**

1, As described in Eq.57, the performance of network $e_{\phi}$ is heavily dependent on the optimality of network v and u. This design may lead to unstable training.

---

> ### Author Response · Authors · 2024-12-04
> **Thanks for your review**
>
> We thank the reviewer for the thoughtful feedback.
>
> **[Weaknesses]**
>
> 1. When we visualized the learned policies of APT, we observed that it remained in an upside-down position while lying on the ground after the pre-training process of the Quadruped. This behavior results in a small state entropy. In contrast, other methods including SEMDICE exhibit much more diverse and dynamic behaviors, such as jumping, getting up, and lying down.
>
> 2. Yes, the code will be publicly available.
>
>
>
> **[Question]**
>
> 1.  As the reviewer mentioned, $e_\phi$ should ideally be optimized using the converged $\nu$ and $\mu$. However, in the context of deep RL, it is common practice to perform alternating gradient updates between two networks rather than waiting for a dependent network to converge. For example, actor-critic algorithms like TD3 alternate between gradient updates for the actor (using the inaccurate Q) and the critic networks, even though the actor update should ideally be based on an accurate $Q^\pi$ estimation. It can be understood that SEMDICE follows a similar approach, and it has been shown to work well in practice.

---

### Official Review · Reviewer_9ykY · 2024-10-28

**Soundness:** 3
**Presentation:** 3
**Contribution:** 3
**Rating:** 6
**Confidence:** 3

**Summary:**

This paper tackles the challenge of unsupervised pre-training in reinforcement learning, aiming to develop a versatile policy that can generalize to downstream tasks without relying on specific reward signals. The authors focus on maximizing state entropy (SEM) to derive a policy that promotes diversity in the agent's experience by increasing the entropy of its stationary state distribution. They introduce **SEMDICE**, a structured off-policy algorithm that derives an SEM policy from arbitrary off-policy data by directly optimizing within the space of stationary distributions. SEMDICE produces a single, stationary policy that effectively maximizes state entropy. Experimental results indicate that SEMDICE surpasses existing baselines in enhancing state entropy and demonstrates superior efficiency in adapting to downstream tasks, making it a strong candidate for SEM-based unsupervised RL pre-training approaches.

**Strengths:**

1. This paper addresses the limitation of existing SEM methods in their inability to support off-policy learning, introducing the objective of optimizing the state distribution. By integrating optimization techniques from the DICE family of methods, the authors propose a well-motivated and logically consistent algorithm.
2. The mathematical derivations in this paper are both accurate and well-founded.  Specifically, this paper provides a detailed explanation of why it is necessary to introduce both $d(s,a)$ and  $\bar{d}(s)$, which is highly convincing from an algorithm design perspective.
3. The experiments demonstrate that the SEMDICE method outperforms all baselines.

In summary, this paper presents a clearly motivated approach, complete algorithmic derivation, and comprehensive experimental analysis.

**Weaknesses:**

1. I think there may be an oversight in the derivation. Specifically, in Equation (15), the summation over $s$ is replaced by sampling $s$ from $D$ through importance sampling, which in fact overlooks cases where $s$ is not in $D$. Therefore,  if $s$ is severely out-of-distribution (OOD), the calculated $ \log \bar{d}(s) $ may be highly inaccurate. Consequently, even though SEMDICE is off-policy, it still introduces a significant bias. The authors might consider adding experiments to demonstrate the impact of OOD-induced bias on the algorithm's performance.

2. In Section 5.1, the different baselines represent various intrinsic reward construction methods, all of which are based on policy-gradient algorithms. Why not use value-based methods as the core algorithms instead? This could help avoid the gradient bias introduced by off-policy policy gradients. Since SEMDICE’s off-policy nature does not suffer from gradient bias, using value-based methods would allow for a fairer comparison between SEMDICE and other algorithms.

**Questions:**

Similar to the issue of weaknesses.

---

> ### Author Response · Authors · 2024-12-04
> **Thanks for your review**
>
> We thank the reviewer for the thoughtful feedback.
>
>
> **[Weaknesses 1]**
>
> As the reviewer pointed out, SEMDICE only uses (s, a, s') samples from the dataset D for optimization. This procedure can be viewed as computing an optimal SEM policy in the maximum-likelihood estimation (MLE) MDP for D. However, this approach is not exclusive to SEMDICE; most off-policy RL algorithms can be understood as the process of computing an optimal policy in an MLE MDP. While it is true that this process is biased, SEMDICE's (in-sample) learning approach encourages the agent to visit more states that it has not previously encountered (see Appendix M for more discussion). As a result, the agent gradually expands the state space it explores and performs well in practice.
>
> For the Monte Carlo integration (line 265), we also tested q(s) = Unif(s), which samples uniformly within the min-max range of each state dimension. However, this caused numerical instability due to the exploding values of $\mu(s)$. Specifically, in Eq. (15), the second term plays a role of reducing $\mu(s)$, while the third term increases it. However, since there is no learning signal to reduce the increased $\mu(s)$ values for OOD states sampled from $s \sim Unif(s)$, $\mu(s)$ easily grows to infinity especially when the state dimensions are large (like Jaco and Quadruped).
>
>
>
> **[Weaknesses 2]**
>
> Following the reviewer's suggestion, we have conducted additional experiments with the value-based baselines in tabular MDPs (see Appendix I in the updated paper). However, please note that the value-based baselines maintain a Q-table rather than an explicit policy, and the target policy is implicitly represented by the Q-values (e.g., greedy for Q or softmax for Q). We have provided both sets of results, but the value-based baselines still significantly underperform SEMDICE. This is expected for the following reasons: (1) For a greedy (deterministic) policy, it is suboptimal because the optimal SEM policy is generally stochastic (see Figure 6 and Proposition 3.1), and (2) for a softmax (stochastic) policy, there is no guarantee that such a naively randomized policy can achieve state entropy maximization.

---

### Official Review · Reviewer_6phi · 2024-11-02

**Soundness:** 3
**Presentation:** 2
**Contribution:** 3
**Rating:** 6
**Confidence:** 4

**Summary:**

This paper focuses on developing an effective pretraining policy in an unsupervised reinforcement learning setting by maximizing state entropy. Previous methods either utilized k-nearest neighbors (kNN)-based particle entropy estimation, which is biased, or were based on the on-policy algorithms from Hazan et al. (2019), which are sampling inefficient. The authors build on the DICE (Stationary Distribution Correction Estimation) series of methods to propose a new off-policy entropy maximization approach named SEMDICE. The paper demonstrates the advantages of their method on benchmarks including Gridworld, MountainCar, and URL Benchmark.

**Strengths:**

The motivation is well. The particle entropy method operates off-policy but optimizes the entropy of the replay buffer rather than the target policy. In contrast, the method proposed by Hazan et al. (2019) is on-policy but sample inefficient. Using DICE-like approaches can address these issues.

While the original DICE framework focuses on Linear MDPs (where the objective function is linear with respect to the state visitation distribution), this paper examines Convex MDPs (where state entropy is convex with respect to the state visitation distribution). This shift introduces differences from the original DICE algorithm, such as the emergence of an exponential term. To mitigate gradient explosion, the paper employs a log-sum-exponential replacement for this term and theoretically demonstrates that this substitution does not significantly impact optimality.

Experimental results indicate that the proposed method, which emphasizes state coverage, outperforms other methods, thereby validating its effectiveness.

**Weaknesses:**

1. The work of Hazan et al. (2019) is a primary reference, yet the paper lacks a comparative analysis of its experiments, particularly like the MountainCar experiment.
2. The implementation details are vague. For instance in Equation 56 of Appendix, the paper estimates $−log d^D(s)$ using a k-nearest neighbors (kNN) based particle entropy estimation, which introduces bias and contradicts the paper's motivation. This needs clarification and should be discussed in the main text.
3. For readers unfamiliar with DICE, the paper is quite opaque. I strongly recommend that the authors organize the theoretical results and the main derivation process similarly to Appendix B in [1], rather than presenting a cumbersome Theorem (e.g., Theorem 3.2).

**Other Typos:**
- Line 49: Missing space before "However."
- Line 296: "inimizing" should be corrected to "minimizing."
- Line 298: The numbering (1), (2) is likely to cause confusion.
- Appendix A: The meaning of $d_+$ is not explained.

Reference：
[1] Dual RL: Unification and New Methods for Reinforcement and Imitation Learning. ICLR 2024.

**Questions:**

1. This paper addresses Convex MDP problems (where state entropy is convex with state visitation distribution). Is there potential for applying the DICE method to other Convex MDP problems, such as diverse skill discovery, as summarized in Table 1 of [2]?
2. The experimental results indicate that the CIC method performs poorly in state coverage during the pretraining phase but excels in downstream task fine-tuning. This suggests that maximizing state entropy may not be the optimal pretraining strategy. Could you provide clarification on this point?

Reference：

[2]  Reward is Enough for Convex MDPs. NeurIPS 2021.

---
Some concerns have been addressed, I have raised my socre.

---

> ### Author Response · Authors · 2024-12-04
> **Thanks for your review**
>
> We thank the reviewer for the thoughtful feedback.
>
>
> **[Weaknesses 1: Comparison with MaxEnt]**
>
> The primary reason we did not compare SEMDICE with MaxEnt (Hazan et al.) in our experiments is that MaxEnt is not designed to compute a single policy but rather to obtain **a set of policies**. Specifically, it maximizes the entropy of the state distribution resulting from the weighted average of state distributions visited by multiple policies. This approach does not align with the scenario we consider, which focuses on fine-tuning a single policy derived from a pre-trained policy. Still, for quantitative comparison with MaxEnt, we have conducted additional experiments in tabular MDPs [1]. In the result, MaxEnt (with tuned hyperparameters) underperforms SEMDICE in terms of learning efficiency. We also would like to emphasize that SEMDICE computes only a single stationary Markov policy, whereas MaxEnt continuously increases the number of policies it maintains, making it computationally inefficient.
>
> [1] https://i.ibb.co/J55vX1j/result-tabular-mdp.png
>
>
>
> **[Weaknesses 2: Particle-based entropy estimation and paper's motivation]**
>
> We would like to clarify that the (approximate) estimation of $-\log d^D(s)$ via kNN does not contradict the motivation of the paper. In Eq. (56), we use Monte Carlo integration to approximate $\log \sum \exp (-\mu(s))$ as $\log \hat{\mathbb{E}}_{d^D} [ \exp( -\mu(s) - \log d^D(s) ) ]$ (see line 265). While this approximation may introduce bias, it still aims to maximize the entropy of the target policy's state stationary distribution from an arbitrary off-policy dataset, rather than maximizing the state entropy of the replay buffer.
>
> In contrast, existing methods focus on maximizing the intrinsic reward $\hat r(s) \approx - \log d^D(s)$, which corresponds to maximizing the state entropy of the replay buffer. That being said, they are approximating $-\log d^\pi(s)$ by $- \log d^D(s)$. As a result, existing methods optimize the **biased objective** even when the kNN approximation error vanishes, whereas SEMDICE optimizes the **unbiased objective** once the kNN approximation error vanishes. We will add this clarification to the paper.
> Please also refer to Appendix H, which shows that SEMDICE can compute the optimal SEM policy even with a dataset collected by a uniform random policy. This behavior of computing an optimal SEM policy from random datasets cannot be achieved by other baselines.
>
>
> **[Weaknesses 3: Paper presentation]**
>
> Thank you for your feedback. We are currently adding more detailed and organized theoretical results and derivations in the Appendix, referring to the mentioned paper, for readers' better understanding. These additions will be reflected in the final version of the paper.
>
>
> **[W: Typos]**
>
> Thanks for pointing out the typos, which are modified in the updated paper.
>
>
>
> **[Questions]**
>
> 1. Yes, we believe that a skill-discovery algorithm (e.g., by maximizing mutual information) can be derived based on the DICE formulation. This would enable MI maximization between tasks and **target policy's states** using an arbitrary off-policy dataset, whereas existing off-policy methods (e.g.  [2])  optimize the mutual information between tasks and **replay buffer's states**. Deriving a DICE-based method for skill-discovery is one of our future directions.
>
>
> 2. We would like to clarify that the optimal pretraining strategy may vary depending on the distribution of downstream tasks. In the case of URLB, only four predefined downstream tasks, designed by humans, are provided for each domain. State entropy maximization approach like SEMDICE would offer a robust initialization that is resilient to worst-case scenarios across a broader range of possible downstream tasks (see Appendix A for more discussion). However, this does not necessarily mean that state entropy maximization is the optimal pretraining strategy for all specific downstream tasks, especially when only a small number of tasks are considered as in URLB. Although CIC's state visitation coverage during pretraining was limited compared to SEMDICE, it can still perform well on certain downstream tasks. To sum up, while state entropy maximization provides a solid foundation for robust performance across diverse tasks, it is not always guaranteed to be the best approach for every specific downstream task.
>
>
> [2] Liu and Abbeel, APS: Active Pretraining with Successor Features, 2021

---

### Official Review · Reviewer_R8AG · 2024-11-04

**Soundness:** 3
**Presentation:** 3
**Contribution:** 3
**Rating:** 6
**Confidence:** 2

**Summary:**

The work proposed a novel state entropy maximization method called SEMDICE for RL pre-training. SEMDICE addresses the limitations of existing methods by directly optimizing the stationary distribution space rather than the policy or Q-function space, utilizing arbitrary offline samples. The study demonstrates that SEMDICE converges to the optimal SEM policy in tabular MDP experiments and is more efficient in RL policy pre-training compared to existing data-driven (i.e., SEM-based) unsupervised RL methods.

**Strengths:**

* The proposed algorithm SEMDICE is a novel method suitable for off-policy training.
* SEMDICE demonstrates efficient convergence in tabular MDP experiments and outperforms existing methods in RL policy pre-training.
* Theoretical analysis is provided to demonstrate the efficacy of the SEMDICE.

**Weaknesses:**

* The hyperparameter $\alpha$ may have a significant impact on the results.
* Though experiments in tabular RL and continuous state spaces are conducted, it does not address high-dimensional data (such as image-based tasks), which may pose computational complexity issues. This could be a potential direction for future improvements.

**Questions:**

* Including the algorithm flowchart within the main text would enhance readers' understanding of the algorithm.
* Is the algorithm sufficiently robust to the hyperparameters, especially the $\alpha$? Is there any method can be employed to mitigate adverse effects?
* How is the computational efficiency in high-dimensional data?

---

> ### Author Response · Authors · 2024-12-04
> **Thanks for your review.**
>
> We thank the reviewer for the thoughtful feedback.
>
> **[Robustnesss to the hyperparameters]**
>
> In Appendix J, we have presented ablation experiments to see the sensitivity of the choice of $\alpha$ and $f$. In short, using too small $\alpha$ incurs numerical instability (due to $\frac{1}{\alpha}$ term in the objective functions), while using too large $\alpha$ can slow down the training efficiency due to its conservative updates. Overall, as can be seen in Figure 10-12, with a moderately large range of the choice of $\alpha \ge 0.1$, SEMDICE exhibits robust performance.
>
>
> **[High-dimensional domains]**
>
> In the paper, we present experimental results on state-based URLB, where the state dimensions are 24 for Walker, 55 for Jaco, and 78 for Quadruped. They provide moderately high-dimensional continuous state space domains, where SEMDICE performs the best in terms of state entropy maximization. Please note that CIC [1] also conducted experiments on state-based URLB only.
>
> Extension of SEMDICE to pixel-based domains remains a future work. Still, although the extension may require combining with an efficient representation learning technique (e.g. [2,3]), SEMDICE would not pose a significant challenge to computational efficiency in high-dimensional domains: SEMDICE optimizes state-dependent function parameterized by two neural networks $\nu_\theta: S \rightarrow \mathbb{R}$ and $\mu_\omega: S \rightarrow \mathbb{R}$ (vs. other methods learn $Q: S \times A \rightarrow \mathbb{R}$ function) with an additional image encoder $E_{\psi}$. Assuming the same encoder network is used as in other methods, the computational cost in training is expected to be similar to that of other methods.
>
>
> **[Algorithm Flow]**
>
> Thanks for your suggestion. We will include an additional simple flowchart to clarify the process. The overall flow of the practical algorithm implementation, along with the pseudo-code, can be found in Appendix G. Optimizing Eq. (58) corresponds to solving the original state entropy maximization problem defined in Eq. (4-6), while Eq. (59-60) describes the policy extraction from the optimized stationary distribution corrections.
>
>
> [1] Laskin et al., CIC: Contrastive Intrinsic Control for Unsupervised Skill Discovery, 2022
>
> [2] Fujimoto et al., A Deep Reinforcement Learning Approach to Marginalized Importance Sampling with the Successor Representation, 2021
>
> [3] Yarats et al., Reinforcement Learning with Prototypical Representations, 2022

---

### Comment · Area_Chair_Bm92 · 2024-11-24
**From AC.**

Dear authors,

If you can, please provide a rebuttal to kick-start the discussion about the paper.

Thanks,

AC

---

### Meta-Review · Area_Chair_Bm92 · 2024-12-20

**Metareview:**

The paper presents an off-policy algorithm that computes the policy with maximal entropy over the state space of the MDP.

The main strengths are: (1) the novelty of the method; (2) good theoretical underpinnings; (3) good empirical performance.

The main weakness is the difficulty is (1) dealing with image / complex observations and (2) a bit opaque writing. I think that these weaknesses are acceptable (you can't do everything in a single paper and the assessment of the writing is subjective).

For this reason, I recommend acceptance.

**Additional Comments On Reviewer Discussion:**

I think this paper represents a solid attempt to do maximum entropy RL and merits consideration for acceptance.

Maxent RL is well within scope for ICLR, the approach seems sensible and the only real shortcoming is dealing with very high dimensional observations. I think this is fine to leave it to future work (you cannot do everything in one paper).

I tried asking reviewers for a discussion, but didn't get any replies. In addition, reviewer qDon provided a low-quality review and failed to fix it on request.

I am reluctantly recommending rejection because I feel I cannot overrule reviewers. I wish I could accept this.

--------------------------------

Update: I have changed the meta-review to recommend acceptance.

---

### Decision · Program_Chairs · 2025-01-22

Accept (Poster)